# Evaluating Fairness Without Sensitive Attributes: A Framework Using Only Auxiliary Models

## Abstract

Although the volume of literature and public attention on machine learning fairness has been growing significantly in recent years, in practice some tasks as basic as measuring fairness, which is the first step in studying and promoting fairness, can be challenging. This is because the sensitive attributes are often unavailable in a machine learning system due to privacy regulations. The straightforward solution is to use auxiliary models to predict the missing sensitive attributes. However, our theoretical analyses show that the estimation error of the directly measured fairness metrics is proportional to the error rates of auxiliary models' predictions. Existing works that attempt to reduce the estimation error often require strong assumptions, e.g. access to the ground-truth sensitive attributes in a subset of samples, auxiliary models' training data and the target data are *i.i.d*, or some form of conditional independence. In this paper, we drop those assumptions and propose a framework that uses *only off-the-shelf auxiliary models*. The main challenge is how to reduce the negative impact of imperfectly predicted sensitive attributes on the fairness metrics without knowing the ground-truth sensitive attribute values. Inspired by the *noisy label learning* literature, we first derive a closed-form relationship between the directly measured fairness metrics and their corresponding ground-truth metrics. And then we estimate some key statistics (most importantly transition matrix in the noisy label literature), which we use, together with the derived relationship, to calibrate the fairness metrics. Our framework can be applied to all popular group fairness definitions as well as multi-class classifiers and multi-category sensitive attributes. In addition, we theoretically prove the upper bound of the estimation error in our calibrated metrics and show our method can substantially decrease the estimation error especially when auxiliary models are inaccurate or the target model is highly biased. Experiments on COMPAS and CelebA validate our theoretical analyses and show our method can measure fairness significantly more accurately than baselines under favorable circumstances.

## 1 Introduction

Despite numerous literature in machine learning fairness (Corbett-Davies & Goel, 2018), in practice even measuring fairness, which is the first step in studying and mitigating fairness, can be challenging as it requires access to sensitive attributes of samples, which are often unavailable due to privacy regulations (Andrus et al., 2021; Holstein et al., 2019; Veale & Binns, 2017). It is a problem that the industry is facing, which significantly slows down the progress of studying and promoting fairness.

Existing methods to estimate fairness without access to ground-truth sensitive attributes mostly fall into two categories. *First*, some methods assume they have access to the ground-truth sensitive attributes on a subset of samples or they can label them if unavailable, *e.g.* Youtube asks its creators to voluntarily provide their demographic information (Wojcicki, 2021). But it either requires labeling resource or depends on the volunteering willingness, and also the resulting measured fairness can be inaccurate due to sampling bias. *Second*, many works assume there exists an auxiliary dataset that can be used to train models to predict the missing sensitive attributes on the target dataset (*i.e.* the dataset that we want to measure the fairness on), *e.g.* Meta (Alao et al., 2021) and others (Elliott et al., 2009; Awasthi et al., 2021; Diana et al., 2022). However, they often need to assume the aux-

iliary dataset and the target dataset are *i.i.d.*, and some form of conditional independence, which are not realistic. In addition, since the auxiliary dataset also contains sensitive information (*i.e.* the sensitive labels), it might be more and more difficult to obtain such training data from the open-source projects given the increasingly stringent privacy regulations today. Note that similar to our work, some researchers also draw insight from noisy label literature (Lamy et al., 2019; Celis et al., 2021; Awasthi et al., 2020). But they assume the noise on sensitive attributes follow assumptions such as conditional independence or known transition probabilities. Furthermore, their goal is to mitigate bias rather than estimating fairness disparity. We emphasize the value of estimating fairness because the metric is vital in reporting and studying fairness in real-world systems.

In this work, we drop many commonly made assumptions, *i.e.* 1) access to labeling resource, 2) access to auxiliary model's training data, 3) data *i.i.d*, and 4) conditional independence. Instead we only rely on *off-the-shelf* auxiliary models, which can be easily obtained via various open-source projects (without their training data). The requirement of the auxiliary model is also flexible. We do not need the auxiliary model's input to share the exactly same feature set as the target data. We only need the auxiliary model's input features have some overlap with the target dataset's features[1]. Our contributions are summarized as follows.

- We theoretically show that directly using auxiliary models to estimate fairness (by predicting the missing sensitive attributes) would lead to a fairness metric whose estimation error is proportional to the prediction error of auxiliary models and the true fairness disparity (Theorem 1, Corollary 1).
- Motivated by the above finding, we propose a general framework (Figure 1, Algorithm 1) to calibrate the noisy fairness metrics using auxiliary models only. The framework is based on a derived closed-form relationship between the directly estimated noisy fairness metrics and their corresponding ground-truth metrics (Theorem 2) in terms of two key statistics: transition matrix and clean prior probability, which are well-studied in the noisy label literature. To estimate them, our framework can leverage any existing estimator. We show an example by adapting HOC (Zhu et al., 2021b) (Algorithm 2). The estimator only assumes that auxiliary models are informative and different auxiliary models make *i.i.d.* predictions.
- We prove the error upper bound of our estimation (Theorem 3), and show that, in a simplified case, our estimated fairness metrics are guaranteed to be closer to the true metrics than the uncalibrated noisy metrics when auxiliary models are inaccurate or the target model is biased (Corollary 2).
- Experiments on COMPAS and CelebA consolidate our theoretical findings and show our calibrated fairness is significantly more accurately than baselines under favorable circumstances.

## 2 PRELIMINARIES

Consider a $K$-class classification problem with *target dataset* $D^\circ := \{(x_n, y_n)|n \in [N]\}$, where $N$ is the number of instances, $x_n$ is the *feature*, and $y_n$ is the *label*. Denote by $\mathcal{X}$ the feature space, $\mathcal{Y} = [K] := \{1, 2, \cdots, K\}$ the label space, and $(X, Y)$ the random variables of $(x_n, y_n), \forall n$. The target model $f : \mathcal{X} \to [K]$ maps $X$ to a predicted label class $f(X) \in [K]$. We aim at measuring group fairness conditioned on a sensitive attribute $A \in [M] := \{1, 2, \cdots, M\}$ which is unavailable in $D^\circ$. Denote the dataset with ground-truth sensitive attributes by $D := \{(x_n, y_n, a_n)|n \in [N]\}$, the joint distribution of $(X, Y, A)$ by $\mathcal{D}$. The task is to estimate the fairness metrics of $f$ on $D^\circ$ *without* sensitive attributes such that the resulting metrics are as close to the fairness metrics evaluated on $D$ (with ground-truth $A$) as possible. See Appendix A.1 for a summary of notations. We consider three group fairness (Wang et al., 2020; Cotter et al., 2019) definitions and their corresponding measurable metrics: *demographic parity* (DP) (Calders et al., 2009; Chouldechova, 2017), *equalized odds* (EOd) (Woodworth et al., 2017), and *equalized opportunity* (EOp) (Hardt et al., 2016).

**Fairness Definitions.** To save space, all our discussions in the main paper are specific to DP. We include the complete derivations for EOd and EOp in the Appendix. DP metric is defined as:

**Definition 1** (Demographic Parity). *The demographic parity metric of $f$ on $\mathcal{D}$ conditioned on $A$ is:*

$$\Delta^{DP}(\mathcal{D}, f) := \frac{1}{M(M-1)K} \sum_{a,a' \in [M], k \in [K]} |\mathbb{P}(f(X) = k|A = a) - \mathbb{P}(f(X) = k|A = a')|.$$

---

[1]For example, if the target dataset contains features about user information (name, location, interests *etc.*), then our method is applicable as long as the auxiliary model can take any one of those features as input and predict sensitive attributes, *e.g.* predicting race from name.

**Matrix-form Metrics.** To unify different fairness metrics, we define matrix $\boldsymbol{H}$ as an intermediate variable. Each column of $\boldsymbol{H}$ denotes the probability needed for evaluating fairness with respect to $f(X)$ (and $Y$ for EOd and EOp). For DP, $\boldsymbol{H}$ is a $M \times K$ matrix. The $a$-th row, $k$-th column, and $(a, k)$-th element of $\boldsymbol{H}$ are denoted by $\boldsymbol{H}[a]$, $\boldsymbol{H}[:,k]$, and $\boldsymbol{H}[a,k]$, respectively. We have $\boldsymbol{H}[:,k] := [\mathbb{P}(f(X) = k|A = 1), \cdots, \mathbb{P}(f(X) = k|A = M)]^\top$. Denote by $\psi(\boldsymbol{H}[a], \boldsymbol{H}[a']) := \|\boldsymbol{H}[a] - \boldsymbol{H}[a']\|_1/\mathsf{col}(\boldsymbol{H})$ the normalized $l_1$ distance between two rows of $\boldsymbol{H}$, where $\mathsf{col}(\boldsymbol{H})$ is the number of columns in $\boldsymbol{H}$. Denote by $\Psi(\boldsymbol{H}) := \sum_{a,\tilde{a}\in[M]} \psi(\boldsymbol{H}[a], \boldsymbol{H}[a'])/(M(M-1))$. We define the following disparity as a general statistical group fairness metric on distribution $\mathcal{D}$ (Chen et al., 2022):

**Definition 2** (Group Fairness Metric). *The group fairness of model $f$ on data distribution $(X, Y, A) \sim \mathcal{D}$ writes as $\Delta(\mathcal{D}, f) = \Psi(\boldsymbol{H})$.*

We can unify Definitions 1, 4, and 5 (in Appendix A.2) using $\boldsymbol{H}$. Next, we study how the fairness metrics can be evaluated without $A$.

**Using Auxiliary Models Directly.** A direct way to measure fairness is to approximate $A$ with an auxiliary model $g : \mathcal{X} \to [M]$ (Ghazimatin et al., 2022; Awasthi et al., 2021; Chen et al., 2019) and get $\widetilde{A} := g(X)$. Note the input of $g$ can be any subsets of feature $X$, and we write the input of $g$ as $X$ just for notation simplicity. In practice, there might be $C$ auxiliary models denoted by the set $\mathcal{G} := \{g_1, \cdots, g_C\}$. The noisy sensitive attributes can be denoted by $\widetilde{A}_c := g_c(X), \forall c \in [C]$ and the corresponding target dataset with $\widetilde{A}$ is $\widetilde{D} := \{(x_n, y_n, (\tilde{a}_n^1, \cdots, \tilde{a}_n^C))|n \in [N]\}$ with its distribution denoted as $\widetilde{\mathcal{D}}$. Similarly, by replacing $A$ with $\widetilde{A}$ in $\boldsymbol{H}$, we can compute $\widetilde{\boldsymbol{H}}$, which is the corresponding matrix-form fairness metric estimated by the auxiliary model $g$ (or $\mathcal{G}$ if multiple auxiliary models are used). Both notations $g$ and $\mathcal{G}$ are used interchangeably in the remainder. Define the directly measured noisy fairness metric of $f$ on $\widetilde{\mathcal{D}}$ as follows.

**Definition 3** (Noisy Group Fairness Metric). *The noisy group fairness of model $f$ on data distribution $(X, Y, \widetilde{A}) \sim \widetilde{\mathcal{D}}$ directly estimated using $g$ writes as $\widetilde{\Delta}(\widetilde{\mathcal{D}}, f) = \Psi(\widetilde{\boldsymbol{H}})$.*

From the above definitions, if we can calibrate the direct noisy estimate $\widetilde{\boldsymbol{H}}$ back to the ground-truth fairness matrix $\boldsymbol{H}$, the estimation error will be greatly reduced. We defer more details to Theorem 2.

**Transition Matrix.** The relationship between $\boldsymbol{H}$ and $\widetilde{\boldsymbol{H}}$ is largely dependent on the relationship between $A$ and $\widetilde{A}$ because it is the single changing variable. Define the matrix $\boldsymbol{T}$ to be the transition probability from $A$ to $\widetilde{A}$ where $(a, \tilde{a})$-th element is $T[a, \tilde{a}] = \mathbb{P}(\widetilde{A} = \tilde{a}|A = a)$. Similarly, denote by $\boldsymbol{T}_k$ the *local* transition matrix conditioned on $f(X) = k$, where the $(a, \tilde{a})$-th element is $T_k[a, \tilde{a}] := \mathbb{P}(\widetilde{A} = \tilde{a}|f(X) = k, A = a)$. Note $\boldsymbol{T}$ can be seen as a *global* transition matrix by weighted averaging $\boldsymbol{T}_k$. Many prior works (Awasthi et al., 2021; Prost et al., 2021; Fogliato et al., 2020) assume $\widetilde{A}$ and $f(X)$ are conditionally independent on $A$. We *drop* this assumption in our theoretical framework. We further define clean (*i.e.* ground-truth) prior probability of $A$ as $\boldsymbol{p} := [\mathbb{P}(A = 1), \cdots, \mathbb{P}(A = M)]^\top$ and the noisy prior probability of $\widetilde{A}$ as $\tilde{\boldsymbol{p}} := [\mathbb{P}(\widetilde{A} = 1), \cdots, \mathbb{P}(\widetilde{A} = M)]^\top$.

## 3 KEY INSIGHT: WHY WE NEED CALIBRATION?

Now we study the error of direct noisy fairness metrics and motivate the necessity of calibration.

**Estimation Error Analysis.** Intuitively, the estimation error of directly measured noisy fairness metrics is dependent on the error of the auxiliary model $g$. Recall $\boldsymbol{p}, \tilde{\boldsymbol{p}}, \boldsymbol{T}$ and $\boldsymbol{T}_k$ are clean prior, noisy prior, global transition matrix, and local transition matrix defined in Sec. 2. Denote by $\boldsymbol{\Lambda}_{\tilde{\boldsymbol{p}}}$ and $\boldsymbol{\Lambda}_{\boldsymbol{p}}$ the square diagonal matrices constructed from $\tilde{\boldsymbol{p}}$ and $\boldsymbol{p}$. We formally prove the upper bound of estimation error in the directly measured metrics in Theorem 1 (See Appendix B.1 for the proof).

**Theorem 1** (Error Upper Bound of Noisy Metrics). *Denote by $\mathsf{Err}^{raw} := |\widetilde{\Delta}^{DP}(\widetilde{\mathcal{D}}, f) - \Delta^{DP}(\mathcal{D}, f)|$ the estimation error of the directly measured noisy fairness metrics. Its upper bound is the following:*

$$\mathsf{Err}^{raw} \leq \frac{2}{K} \sum_{k \in [K]} \Big( \bar{h}_k \underbrace{\|\boldsymbol{\Lambda}_{\tilde{\boldsymbol{p}}}(\boldsymbol{T}^{-1}\boldsymbol{T}_k - \boldsymbol{I})\boldsymbol{\Lambda}_{\tilde{\boldsymbol{p}}}^{-1}\|_1}_{cond.\ indep.\ violation} + \delta_k \underbrace{\|\boldsymbol{\Lambda}_{\boldsymbol{p}}\boldsymbol{T}_k\boldsymbol{\Lambda}_{\tilde{\boldsymbol{p}}}^{-1} - \boldsymbol{I}\|_1}_{error\ of\ g} \Big),$$

*where $\bar{h}_k := \frac{1}{M} \sum_{a \in [M]} H[a, k]$, $\delta_k := \max_{a \in [M]} |H[a, k] - \bar{h}_k|$.*

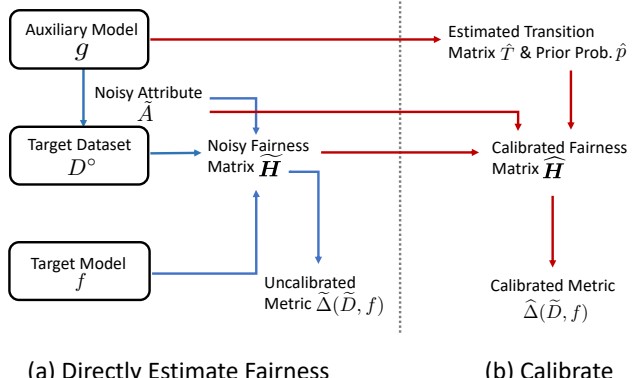

Figure 1: Overview of our framework to estimate fairness without sensitive attribute given auxiliary models $g$ only. Step 1 (a): Estimate noisy fairness matrix. Step 2 (b): Calibrate the fairness matrix using the estimated transition matrix and prior probability.

Theorem 1 reveals that the estimation error of directly measured metric depends on:

- $\bar{h}_k$: The average confidence of $f(X)$ on class $k$ over all sensitive groups. For example, if $f$ is a crime prediction model and $A$ is race, a biased $f$ (Angwin et al., 2016) may predict that the crime ($k = 1$) rate for different races are 0.1, 0.2 and 0.6 respectively, then $\bar{h}_1 = \frac{0.1+0.2+0.6}{3} = 0.3$, and it is an approximation (unweighted by sample size) of the average crime rate over the entire population. The term is dependent on $\mathcal{D}$ and $f$, and independent of any estimation algorithm.
- $\delta_k$: The maximum disparity between confidence of $f(X)$ on class $k$ and average confidence $\bar{h}_k$ across all sensitive groups. Using the same example, $\delta_1 = \max(|0.1 - 0.3|, |0.2 - 0.3|, |0.6 - 0.3|) = 0.3$. It is an approximation of the underlying fairness disparity, and larger $\delta_k$ indicates $f$ is more biased on $\mathcal{D}$. The term is also dependent on $\mathcal{D}$ and $f$ (i.e. the true fairness disparity), and independent of any estimation algorithm.
- *Conditional Independence Violation*: The term is dependent on the auxiliary model $g$'s prediction $\tilde{A}$ in terms of the transition matrix ($\boldsymbol{T}$ and $\boldsymbol{T}_k$) and noisy prior probability ($\tilde{\boldsymbol{p}}$). The term goes to 0 when $\boldsymbol{T} = \boldsymbol{T}_k$, which implies $\tilde{A}$ and $f(X)$ are independent conditioned on $A$. This is the common assumption made in the prior work (Awasthi et al., 2021; Prost et al., 2021; Fogliato et al., 2020). And this term measures how much the conditional independence assumption is violated.
- *Error of $g$*: Similarly, this term is dependent on the auxiliary model $g$. It goes to 0 when $\boldsymbol{T}_k = \boldsymbol{I}$ which implies the error rates of $g$'s prediction is 0, i.e. $g$ is perfectly accurate. It measures the impact of $g$'s error on the fairness estimation error.

To help better understand the upper bound, we consider a simplified case when $f$ is a binary model and $A$ is a binary variable. We further assume the conditional independence condition to remove the third term listed above in Theorem 1. See Appendix A.3 for the formal definition of conditional independence.[2] Corollary 1 summarizes the result.

**Corollary 1.** *For a binary classifier $f$ and a binary sensitive attribute $A \in \{1, 2\}$, when $(\tilde{A} \perp\!\!\!\perp f(X)|A)$ holds, Theorem 1 is simplified to $\mathsf{Err}^{raw} \leq 2\delta(e_1 + e_2)$, where $e_1$ and $e_2$ are transition probabilities from noisy attributes to clean attributes,* i.e. $e_1 = \mathbb{P}(A = 1|\tilde{A} = 2)$, $e_2 = \mathbb{P}(A = 2|\tilde{A} = 1)$, $\delta = |\mathbb{P}(f(X) = 1|A = 1) - \mathbb{P}(f(X) = 1|A = 2)|/2$.

**Why Calibrate?**  Corollary 1 clearly shows the estimation error of the directly measured fairness is proportional to the true underlying disparity between sensitive groups (i.e. $\delta$) and the auxiliary model's error rates (i.e. $e_1$ and $e_2$). In other words, the uncalibrated metrics can be highly inaccurate when $f$ is highly biased or $g$ has poor performance. Both are practical cases since when we want to measure $f$'s fairness, it has already shown some fairness-related concerns and the fairness disparity is not negligible. Moreover, the auxiliary model $g$ is usually not highly accurate due to distribution shift. Hence, in those cases we should calibrate the metrics to get more accurate measurements.

---

[2]We only assume it for the purpose of demonstrating a less complicated theoretical result, we do not need this assumption in our proposed algorithm later.

## 4 METHODOLOGY

In this section, we introduce our calibration framework and algorithm (Sec. 4.1), prove the error upper bounds for our calibration methods (Sec. 4.2), and elaborate key steps of our algorithm (Sec. 4.3).

### 4.1 PROPOSED FRAMEWORK

With a given auxiliary model $g$ that labels sensitive attributes, we can anatomize the relationship between the true disparity and the noisy disparity. We have the following theorem for DP. See Appendix B.2 for results with respect to EOd and EOp and their proofs.

**Theorem 2.** *[Closed-form Relationship (DP)] The closed-form relationship between the true fairness vector $\boldsymbol{H}[:,k]$ and the corresponding directly measured noisy fairness vector $\widetilde{\boldsymbol{H}}[:,k]$ is the following:*

$$\boldsymbol{H}[:,k] = (\boldsymbol{T}_k^\top \boldsymbol{\Lambda_p})^{-1} \boldsymbol{\Lambda_{\tilde{p}}} \widetilde{\boldsymbol{H}}[:,k], \forall k \in [K].$$

**Framework Overview.** Theorem 2 reveals that the noisy disparity and the corresponding true disparity are related in terms of three key statistics: noisy prior $\tilde{\boldsymbol{p}}$, clean prior $\boldsymbol{p}$, and local transition matrix $\boldsymbol{T}_k$. Ideally, if we can obtain the ground-truth values of them, then we can calibrate the noisy fairness vectors to their corresponding ground-truth vectors (and therefore the perfectly accurate fairness metrics) using the closed-form in Theorem 2. Hence, the most important step is to estimate $\boldsymbol{T}_k$, $\boldsymbol{p}$, and $\tilde{\boldsymbol{p}}$ without knowing the ground-truth values of $A$. Once we have those estimated key statistics, we can easily plug them into the above equation as the calibration step. Figure 1 shows the overview of our framework.

**Algorithm.** We summarize our framework in Algorithm 1. In Line 4, we use sample mean in the uncalibrated form to estimate $\widetilde{\boldsymbol{H}}$ as $\widetilde{H}[\tilde{a}, k] = \mathbb{P}(f(X) = k | \widetilde{A} = \tilde{a}) \approx \frac{1}{N} \sum_{n=1}^N \mathbb{1}(f(x_n = k | \tilde{a}_n = \tilde{a}))$ and $\tilde{\boldsymbol{p}}$ as $\tilde{\boldsymbol{p}}[\tilde{a}] = \mathbb{P}(\widetilde{A} = \tilde{a}) \approx \frac{1}{N} \sum_{n=1}^N \mathbb{1}(\tilde{a}_n = \tilde{a}), \forall \tilde{a} \in [M]$. In Line 6, we plug in an existing transition matrix and prior probability estimator to estimate $\boldsymbol{T}_k$ and $\boldsymbol{p}$ with only mild adaption that will be introduced in Sec. 4.3. Note that although we choose a specific estimator to use, our framework is flexible and compatible with any `StatEstimator` proposed in the noisy label literature (Liu & Chen, 2017; Zhu et al., 2021b; 2022).

### 4.2 ESTIMATION ERROR ANALYSIS

We theoretically analyze estimation error on our calibrated metrics in a similar way as in Sec. 3. The derivation is based on local estimates $\widehat{\boldsymbol{T}}_k$, global estimates $\widehat{\boldsymbol{T}}$ would be similar. Denote by $\widehat{\Delta}^{\mathsf{DP}}(\widetilde{\mathcal{D}}, f)$ the calibrated DP disparity evaluated on our calibrated fairness matrix $\widehat{\boldsymbol{H}}$. We have:

**Theorem 3** (Error Upper Bound of Calibrated Metrics). *Denote the estimation error of the calibrated fairness metrics by $\mathsf{Err}^{cal} := |\widehat{\Delta}^{\mathsf{DP}}(\widetilde{\mathcal{D}}, f) - \Delta^{\mathsf{DP}}(\mathcal{D}, f)|$. Its upper bound is the following:*

$$\mathsf{Err}^{cal} \leq \frac{2}{K} \sum_{k \in [K]} \left\| \boldsymbol{\Lambda_p}^{-1} \right\|_1 \left\| \boldsymbol{\Lambda_p} \boldsymbol{H}[:,k] \right\|_\infty \varepsilon(\widehat{\boldsymbol{T}}_k, \hat{\boldsymbol{p}}),$$

*where $\varepsilon(\widehat{\boldsymbol{T}}_k, \hat{\boldsymbol{p}}) := \|\boldsymbol{\Lambda_{\hat{p}}}^{-1} \boldsymbol{\Lambda_p} - \boldsymbol{I}\|_1 \|\boldsymbol{T}_k \widehat{\boldsymbol{T}}_k^{-1}\|_1 + \|\boldsymbol{I} - \boldsymbol{T}_k \widehat{\boldsymbol{T}}_k^{-1}\|_1$ is the error induced by calibration. With a perfect estimator, i.e. $\widehat{\boldsymbol{T}}_k = \boldsymbol{T}_k$ and $\hat{\boldsymbol{p}}_k = \boldsymbol{p}_k, \forall k \in [K]$, we have $\mathsf{Err}^{cal} = 0$.*

Theorem 3 shows the upper bound of estimation error mainly depends on the estimates $\widehat{\boldsymbol{T}}_k$ and $\hat{\boldsymbol{p}}$, i.e., the following two terms in $\varepsilon(\widehat{\boldsymbol{T}}_k, \hat{\boldsymbol{p}})$: $\|\boldsymbol{\Lambda_{\hat{p}}}^{-1} \boldsymbol{\Lambda_p} - \boldsymbol{I}\|_1 \|\boldsymbol{T}_k \widehat{\boldsymbol{T}}_k^{-1}\|_1$ and $\|\boldsymbol{I} - \boldsymbol{T}_k \widehat{\boldsymbol{T}}_k^{-1}\|_1$. When the estimates are perfect, *i.e.* $\widehat{\boldsymbol{T}}_k = \boldsymbol{T}_k$ and $\hat{\boldsymbol{p}} = \boldsymbol{p}$, then both terms go to 0 because $\boldsymbol{\Lambda_{\hat{p}}}^{-1} \boldsymbol{\Lambda_p} = \boldsymbol{I}$ and $\boldsymbol{T}_k \widehat{\boldsymbol{T}}_k^{-1} = \boldsymbol{I}$. We now compare the above error upper bound with the exact error (not its upper bond) in the case of Corollary 1, and summarize the result in Corollary 2.

**Corollary 2.** *When assumptions in Corollary 1 hold, further assume $\boldsymbol{p} = [0.5, 0.5]^\top$, then the proposed calibration method is guaranteed to be more accurate than the uncalibrated measurement, i.e., $\mathsf{Err}^{cal} \leq \mathsf{Err}^{raw}$, if $\varepsilon(\widehat{\boldsymbol{T}}_k, \hat{\boldsymbol{p}}) \leq \gamma := \max_{k' \in \{1,2\}} \frac{e_1 + e_2}{1 + \frac{\|\boldsymbol{H}[:,k']\|_1}{\Delta^{\mathsf{DP}}(\mathcal{D}, f)}}, \forall k \in \{1, 2\}.$*

---

**Algorithm 1** Fairness calibration framework ($\mathsf{DP}$)

---

1: **Input:** A set of auxiliary models $\mathcal{G} = \{g_1, \cdots, g_C\}$. Target dataset $D^\circ$. Target model $f$. Transition matrix and prior probability estimator $\mathtt{StatEstimator}$.

2: $\tilde{a}_n^c \leftarrow g_c(x_n), \forall c \in [C], n \in [N]$         *# Predict sensitive attributes using all $g \in \mathcal{G}$*

3: $\widetilde{D} \leftarrow \{(x_n, y_n, (\tilde{a}_n^1, \cdots, \tilde{a}_n^C)) | n \in [N]\}$     *# Build the dataset with noisy sensitive attributes*

4: $\widetilde{\boldsymbol{H}}, \tilde{\boldsymbol{p}} \leftarrow \mathtt{DirectEst}(\widetilde{D}, f)$       *# Directly estimate fairness matrix and prior with sample mean*

5: *# Estimate key statistics: $\boldsymbol{p}$ and $\boldsymbol{T}_k$*

6: $\{\widehat{\boldsymbol{T}}_1, \cdots, \widehat{\boldsymbol{T}}_K\}, \hat{\boldsymbol{p}} \leftarrow \mathtt{StatEstimator}(\widetilde{D}, f)$

7: $\forall k \in [K] : \widehat{\boldsymbol{H}}[:,k] \leftarrow (\widehat{\boldsymbol{T}}_k^\top \boldsymbol{\Lambda}_{\hat{\boldsymbol{p}}})^{-1} \boldsymbol{\Lambda}_{\tilde{\boldsymbol{p}}} \widetilde{\boldsymbol{H}}[:,k]$      *# Calibrate each fairness vector with Theorem 2*

8: $\widehat{\Delta}(\widetilde{D}, f) \leftarrow \Psi(\widehat{\boldsymbol{H}})$          *# Calculate the final fairness metric as Definition 2*

9: **Output:** The calibrated fairness metric $\widehat{\Delta}(\widetilde{D}, f)$

---

Corollary 2 shows when the error $\varepsilon(\widehat{\boldsymbol{T}}_k, \hat{\boldsymbol{p}})$ that is induced by inaccurate $\widehat{\boldsymbol{T}}_k$ and $\hat{\boldsymbol{p}}$ is below the threshold $\gamma$, our method is guaranteed to lead to a smaller estimation error compared to the uncalibrated measurement under the considered setting. The threshold implies that, adopting our method rather than the uncalibrated measurement can be greatly beneficial when $e_1$ and $e_2$ are high (*i.e.* $g$ is inaccurate) or when the normalized (true) fairness disparity $\frac{\Delta^{\mathsf{DP}}(\mathcal{D}, f)}{\|\boldsymbol{H}[:,k']\|_1}$ is high (*i.e.* $f$ is highly biased).

## 4.3 ESTIMATING KEY STATISTICS

As mentioned, our framework can plug in any existing estimator of transition matrix and prior probability. We choose HOC (Zhu et al., 2021b) because it is free of training. Some methods (Liu & Tao, 2015; Scott, 2015; Patrini et al., 2017) require extra training with target data and auxiliary model outputs, which introduces extra cost. Moreover, it brings a practical challenge in hyperparameter tuning given we have no ground-truth sensitive attributes. HOC decodes $\boldsymbol{T}_k$ by checking both the agreements and disagreements among noisy attributes (auxiliary model predictions). See more details in Appendix C.1. For a successful decoding, HOC makes the following assumptions:

**Assumption 1** (HOC: Informativeness). *The noisy attributes given by each classifier $g$ are informative, i.e. $\forall k \in [M]$, 1) $T_k$ is non-singular and 2) either $T_k[a, a] > \mathbb{P}(\widetilde{A} = a | f(X) = k)$ or $T_k[a, a] > T_k[a, a'], \forall a' \neq a$.*

**Assumption 2** (HOC: Independence). *Given three auxiliary models, the noisy attributes predicted by them are independent and identically distributed (i.i.d.), i.e., $g_1(X), g_2(X),$ and $g_3(X)$ are i.i.d.*

Assumption 1 is the prerequisite of getting a feasible and unique estimate of $\boldsymbol{T}_k$ (Zhu et al., 2021b), where the non-singular assumption ensures the matrix inverse in Theorem 2 exists and the constraints on $T_k[a, a]$ describes the worst tolerable performance of $g$. When $M = 2$, the constraints can be simplified as $T_k[1, 2] + T_k[2, 1] < 1$ (Liu & Chen, 2017; Liu & Guo, 2020). If this assumption is violated, there might exist more than one feasible estimates of $\boldsymbol{T}_k$, making the problem insoluble. Assumption 2 ensures the additional two auxiliary models provide more information than using only one classifier. Note it has been proved by Liu (2022) that *three* is the sufficient and necessary number of auxiliary models to provide sufficient information to identify $\boldsymbol{T}_k$. If Assumption 2 is violated, we would still get an estimate but may be inaccurate.

**Adapting HOC.** Algorithm 2 shows how we adapt HOC as $\mathtt{StatEstimator}$ (in Algorithm 1, Line 6), namely $\mathtt{HOCFair}$. The original HOC uses one auxiliary model and simulates the other two based on clusterability assumption (Zhu et al., 2021b), which assumes $x_n$ and its 2-nearest-neighbors share the same true sensitive attribute, and therefore their noisy attributes can be used to simulate the output of auxiliary models. If this assumption does not hold (Zhu et al., 2022), we can directly use more auxiliary models. With a sufficient number of noisy attributes, we can randomly select *three* of them for every sample as Line 6, and then approximate $\boldsymbol{T}_k$ with $\widehat{\boldsymbol{T}}_k$ in Line 8. In our experiments, we test both using one auxiliary model and multiple auxiliary models.

---

**Algorithm 2** StatEstimator: HOCFair

---

1: **Input:** Noisy dataset $\widetilde{D}$. Target model $f$.
2: $C \leftarrow \texttt{\#Attribute}(\widetilde{D})$        *# Get the number of noisy attributes (i.e. number of aux. models)*
3: **if** $C < 3$ **then**       *# Get 2-Nearest-Neighbors of $x_n$ and save their attributes as $x_n$'s attribute*
4:     $\widetilde{D} \leftarrow \{(x_n, y_n, (\tilde{a}_n^1, \cdots, \tilde{a}_n^{3C}))|n \in [N]\} \leftarrow \texttt{Get2NN}(\widetilde{D})$
5: **end if**
6: $\{(\tilde{a}_n^1, \tilde{a}_n^2, \tilde{a}_n^3)|n \in [N]\} \leftarrow \texttt{Sample}(\widetilde{D})$     *# Randomly sample 3 noisy attributes for each instance*
7: $(\widehat{\boldsymbol{T}}, \hat{\boldsymbol{p}}) \leftarrow \textsf{HOC}(\{(\tilde{a}_n^1, \tilde{a}_n^2, \tilde{a}_n^3)|n \in [N]\})$     *# Use HOC to get global estimates $\boldsymbol{T} \approx \widehat{\boldsymbol{T}}$ and $\boldsymbol{p} \approx \hat{\boldsymbol{p}}$*
8: $(\widehat{\boldsymbol{T}}_k, -) \leftarrow \textsf{HOC}(\{(\tilde{a}_n^1, \tilde{a}_n^2, \tilde{a}_n^3)|n \in [N], f(x_n) = k\}), \ \forall k \in [K]$ *# Get local estimates $\boldsymbol{T}_k \approx \widehat{\boldsymbol{T}}_k$*
9: **Output:** $\{\widehat{\boldsymbol{T}}_1, \cdots, \widehat{\boldsymbol{T}}_K\}, \hat{\boldsymbol{p}}$     *# Return the estimated statistics*

---

## 5 EXPERIMENTS

### 5.1 EXPERIMENT SETUP

We test the performance of our method on two real-world datasets: COMPAS (Angwin et al., 2016) and CelabA (Liu et al., 2015). We report results on all three group fairness metrics (DP, EOd, and EOp) whose true disparities (estimated using the ground-truth sensitive attributes) are denoted by $\Delta^{\textsf{DP}}(\mathcal{D}, f), \Delta^{\textsf{EOd}}(\mathcal{D}, f), \Delta^{\textsf{EOp}}(\mathcal{D}, f)$ respectively. We train the target model $f$ on the dataset without using $A$, and use the auxiliary models downloaded from open-source projects. The detailed settings are the following:

- **COMPAS** (Angwin et al., 2016): Recidivism prediction data. Feature $X$: tabular data. Label $Y$: recidivism within two years (binary). Sensitive attribute $A$: race (black and non-black). Target models $f$ (trained by us): decision tree, random forest, boosting, SVM, logit model, and neural network (accuracy range 66%–70% for all models). Three auxiliary models $(g_1, g_2, g_3)$: racial classifiers given name as input Sood & Laohaprapanon (2018) (average accuracy 68.85%).
- **CelabA** (Liu et al., 2015): Face dataset. Feature $X$: facial images. Label $Y$: smile or not (binary). Sensitive attribute $A$: gender (male and female). Target models $f$: ResNet18 (He et al., 2016) (accuracy 90.75%, trained by us). We only use one auxiliary model $(g_1)$: gender classifier that takes facial images as input (Serengil & Ozpinar, 2021) (accuracy 92.55%). We then use the clusterability to simulate the other two auxiliary models as Line 3 in Algorithm 2.

**Practical Estimates of $\boldsymbol{T}_k$: Local vs. Global.** According to Theorem 3, when $\boldsymbol{T}_k$s are accurately estimated, we should always rely on the local estimates as Line 8 of Algorithm 2 to achieve a zero calibration error. However, in practice, each time when we estimate a local $\widehat{\boldsymbol{T}}_k$, the estimator would introduce certain error on the $\widehat{\boldsymbol{T}}_k$ (discussed in Sec. 4.3) and the matrix inversion in Theorem 2 might amplify the estimation error on $\widehat{\boldsymbol{T}}_k$ each time, leading to a large overall error on the metric. One *heuristic* is to use a single global transition matrix $\widehat{\boldsymbol{T}}$ estimated once on the full dataset $\widetilde{D}$ as Line 7 of Algorithm 2 to replace all $\widehat{\boldsymbol{T}}_k$'s. Intuitively, $\widehat{\boldsymbol{T}}$ can be viewed as the weighted average of all $\widehat{\boldsymbol{T}}_k$'s to stabilize estimation error (variance reduction) on $\widehat{\boldsymbol{T}}_k$. Admittedly, the average will introduce bias since the equation in Theorem 2 would not hold when replacing $\boldsymbol{T}_k$ with $\boldsymbol{T}$. The justification is that the error introduced by violating the equality might be smaller than the error introduced by using severely inaccurately estimates of $\boldsymbol{T}_k$'s. Therefore, we offer two options for estimating $\boldsymbol{T}_k$ in practice: locals estimates $\boldsymbol{T}_k \approx \widehat{\boldsymbol{T}}_k$ and global estimates $\boldsymbol{T}_k \approx \widehat{\boldsymbol{T}}$. Although it is hard to guarantee which option must be better in reality, we report the experimental results using both options and provide insights for choosing between both estimates in Sec. 5.2.

**Method.** We test our proposed framework with global estimates $\boldsymbol{T}_k \approx \widehat{\boldsymbol{T}}$ (Global) and local estimates $\boldsymbol{T}_k \approx \widehat{\boldsymbol{T}}_k$ (Local). We compare with two baselines: the directly estimated metric without any calibration (Base) and Soft (Chen et al., 2019) which also only uses auxiliary models to calibrate the measured fairness by re-weighting metric with the soft predicted probability from the auxiliary model.

Table 1: Normalized estimation error on COMPAS. Each row represents a different target model $f$.

| COMPAS
*True disparity:* $\sim 0.2$ | DP *Normalized Error (%)* $\downarrow$ | | | | EOd *Normalized Error (%)* $\downarrow$ | | | | EOp *Normalized Error (%)* $\downarrow$ | | | |
|---|---|---|---|---|---|---|---|---|---|---|---|---|
| | Base | Soft | Global | Local | Base | Soft | Global | Local | Base | Soft | Global | Local |
| tree | 43.82 | 61.26 | **22.29** | 39.81 | 45.86 | 63.96 | **23.09** | 42.81 | 54.36 | 70.15 | **13.27** | 49.49 |
| forest | 43.68 | 60.30 | **19.65** | 44.14 | 45.60 | 62.85 | **18.56** | 44.04 | 53.83 | 69.39 | **17.51** | 63.62 |
| boosting | 43.82 | 61.26 | **22.29** | 44.64 | 45.86 | 63.96 | **23.25** | 49.08 | 54.36 | 70.15 | **13.11** | 54.67 |
| SVM | 50.61 | 66.50 | **30.95** | 42.00 | 53.72 | 69.69 | **32.46** | 47.39 | 59.70 | 71.12 | **29.29** | 51.31 |
| logit | 41.54 | 60.78 | **16.98** | 35.69 | 43.26 | 63.15 | **21.42** | 31.91 | 50.86 | 65.04 | **14.90** | 26.27 |
| nn | 41.69 | 60.55 | **19.48** | 34.22 | 43.34 | 62.99 | **19.30** | 43.24 | 54.50 | 68.50 | **14.20** | 59.95 |
| compas_score | 41.28 | 58.34 | **11.24** | 14.66 | 42.43 | 59.79 | **11.80** | 18.65 | 48.78 | 62.24 | **5.78** | 23.80 |

Table 2: Normalized error on CelebA. Each row represents a different pre-trained model to generate feature representations that we use to simulate the other two auxiliary models $g_2$, $g_3$ (Line 3, Algorithm 2). Base and Soft are computed on $g_1$ and *not changed* since they are independent of feature representations. The ground-truth fairness metrics are DP: 0.13, EOd: 0.03, EOp: 0.05.

| CelebA | DP *Normalized Error (%)* $\downarrow$ | | | | EOd *Normalized Error (%)* $\downarrow$ | | | | EOp *Normalized Error (%)* $\downarrow$ | | | |
|---|---|---|---|---|---|---|---|---|---|---|---|---|
| | Base | Soft | Global | Local | Base | Soft | Global | Local | Base | Soft | Global | Local |
| Facenet | 15.33 | 12.54 | 22.17 | **10.89** | 4.11 | 6.46 | 7.54 | **0.26** | 2.82 | **0.34** | 12.22 | 2.93 |
| Facenet512 | 15.33 | 12.54 | 21.70 | **7.26** | 4.11 | 6.46 | 4.85 | **0.52** | 2.82 | **0.34** | 11.80 | 3.24 |
| OpenFace | 15.33 | 12.54 | 10.31 | **9.39** | **4.11** | 6.46 | 10.43 | 5.03 | 2.82 | **0.34** | 0.56 | 0.93 |
| ArcFace | 15.33 | 12.54 | 19.59 | **9.69** | 4.11 | 6.46 | 5.72 | **0.23** | 2.82 | **0.34** | 11.16 | 3.85 |
| Dlib | 15.33 | 12.54 | 15.09 | **5.30** | **4.11** | 6.46 | 4.87 | 4.25 | 2.82 | **0.34** | 9.74 | 2.32 |
| SFace | 15.33 | 12.54 | 17.00 | **4.77** | 4.11 | 6.46 | 4.04 | **3.91** | 2.82 | **0.34** | 9.36 | 3.28 |

**Evaluation Metric.** Let $\Delta(D, f)$ be the ground-truth fairness metric. For a given estimated metric $E$, we define three estimation errors: Raw Error$(E) := |E - \Delta(D, f)|$, Normalized Error$(E) := \frac{\text{Raw Error(E)}}{\Delta(D, f)}$, and Improvement$(E) := 1 - \frac{\text{Raw Error(E)}}{\text{Raw Error(Base)}}$ where Base is the directly measured metric.

## 5.2 RESULTS AND ANALYSES

**COMPAS Results.** Table 1 reports the normalized error on COMPAS (See Table 7 in Appendix D.1 for the other two evaluation metrics). There are two main observations. *First*, our calibrated metrics outperform baselines with a big margin on all three fairness definitions. Compared to Base, our metrics are 39.6%–88.2% more accurate (Improvement). As pointed out by Corollary 2, this is because the target models $f$ are highly biased (Table 6) and the auxiliary models $g$ are inaccurate (accuracy 68.9%). As a result, Base has large normalized error (40–60%). *Second*, Global outperforms Local, since with inaccurate auxiliary models, Assumptions 1–2 on HOC estimator may not hold in local dataset, inducing large estimation errors in local estimates.

**CelebA Results.** Table 2 reports the normalized error on CelebA where each row represents using a different pre-trained model to generate feature representations used to simulate the other two auxiliary models (See Table 8 in Appendix D.2 for the full results). We have two observations. *First*, although our method still outperforms baselines most of time, the margin is smaller and we are underperformed by Soft when estimating EOp. Similarly this is because the conditions in Corollary 2 do not hold, *i.e.* $f$ is barely biased in EOd and EOp (Table 8) and $g$ is accurate (accuracy 92.6%). As a result, Base only has a moderate normalized error in DP (15.3%), and small normalized errors in EOd (4.1%) and EOp (2.8%). Given the highly accurate Base, the benefit of adapting our method is outweighed by the estimation error introduced by calibration (mostly the key statistic estimator). *Second*, contrary to COMPAS, Local outperforms Global. This is because now the auxiliary models are accurate, Assumption 1 always holds and Assumption 2 is also likely to hold when $g_2$ and $g_3$ are well-simulated. Consequently, Local is estimated accurately while Global induces extra error due to violating the equality in Theorem 2. Finally, even though our method is underperformed in EOp, the raw error of our method is acceptable, which is less than 0.01 as Table 8 in Appendix D.2.

**Ablation Study.** To better understand when our method can give a clear advantage with different quality of $g$, we run an ablation study on CelebA. We randomly flip the predicted sensitive attributes by $g$ to bring down $g$'s accuracy and report the results in Table 3 (See Appendix D.2 for the full results). When $g$ becomes less accurate, our method can outperform baselines, which validates

Table 3: Normalized error on the CelebA when adding noise by randomly flipping the predicted attributes to bring down the performance of auxiliary models. Each row represents the noise magnitude and accuracy of auxiliary models, *e.g.* "[0.2, 0.0] (82.44%)" means $T[1, 2] = 0.2$, $T[2, 1] = 0.0$ and accuracy is 82.44%.

| CelebA FaceNet512 | DP *Normalized Error (%)* ↓ | | | | EOd *Normalized Error (%)* ↓ | | | | EOp *Normalized Error (%)* ↓ | | | |
|---|---|---|---|---|---|---|---|---|---|---|---|---|
| | Base | Soft | Global | Local | Base | Soft | Global | Local | Base | Soft | Global | Local |
| [0.2, 0.0] (82.44%) | 7.37 | 11.65 | 20.58 | **5.05** | 25.06 | 26.99 | 6.43 | **0.10** | 24.69 | 27.27 | 11.11 | **1.07** |
| [0.2, 0.2] (75.54%) | 30.21 | 31.57 | 24.25 | **13.10** | 44.73 | 46.36 | 11.26 | **9.04** | 37.67 | 38.77 | **20.94** | 27.98 |
| [0.4, 0.2] (65.36%) | 51.32 | 54.56 | 19.42 | **10.47** | 62.90 | 65.10 | **11.09** | 19.15 | 56.51 | 58.73 | 23.86 | **23.55** |
| [0.4, 0.4] (58.45%) | 77.76 | 78.39 | **9.41** | 19.80 | 79.31 | 80.10 | 24.49 | **8.02** | 78.35 | 79.62 | 10.61 | **5.71** |

Corollary 2. In addition, Local still outperforms Global. This is because we add random noise following Assumption 2 and therefore the estimation error of Local is not increased significantly.

**Takeaways.** Our experimental results imply two takeaways: 1) our calibration method can give a clear advantage when the error rates of $g$ are moderate to high (*e.g.* error $\geq 15\%$) or $f$ is highly biased (*e.g.* fairness disparity $\geq 0.1$) and 2) we can prefer Local when the auxiliary model is accurate and Global otherwise. In practice, given no ground-truth $A$, we can roughly estimate auxiliary models' accuracy range from the estimated transition matrix $\widehat{T}$.

## 6 RELATED WORK

**Fairness with Imperfect Sensitive Attributes.** The closest work to ours is (Chen et al., 2019), which also assumes only auxiliary models. It is only applicable to *demographic disparity*, and we compare it in the experiments. In addition, other works focus on how to train auxiliary models from a given auxiliary dataset (Awasthi et al., 2021; Diana et al., 2022). For example, Awasthi et al. (2021) propose an active learning scheme and assume there exists an auxiliary dataset that is *i.i.d* with the target dataset. In our work, we do not need the auxiliary dataset; nor do we need to assume the auxiliary model's training set is *i.i.d* with the target dataset. Lamy et al. (2019) also draws insights from noisy label literature. However, the attribute noise is assumed to come from the mutually contaminated assumption rather than from an auxiliary model. Furthermore, Prost et al. (2021) and Fogliato et al. (2020) theoretically study the error gap of estimating fairness, but they do not propose any calibration method. There are other parallel works that aim to mitigate bias without estimating it (Hashimoto et al., 2018; Lahoti et al., 2020; Wang et al., 2020; Yan et al., 2020). We emphasize the value of estimating fairness because the metrics themselves are vital in reporting and studying fairness in real-world systems.

**Noisy Label Learning.** Label noise may come from various sources, e.g., human annotation error (Xiao et al., 2015; Wei et al., 2022; Agarwal et al., 2016) and model prediction error (Lee et al., 2013; Berthelot et al., 2019; Zhu et al., 2021a), which can be characterized by transition matrix on label (Liu, 2022; Bae et al., 2022; Yang et al., 2021). Applying the noise transition matrix to ensure fairness is emerging (Wang et al., 2021; Liu & Wang, 2021; Lamy et al., 2019). There exist two lines of works for estimating transition matrix. The first line relies on anchor points (samples belonging to a class with high certainty) or their approximations (Liu & Tao, 2015; Scott, 2015; Patrini et al., 2017; Xia et al., 2019; Northcutt et al., 2021). These works requires training a neural network on the $(X, \widetilde{A} := g(X))$. The second line of work, which we leverage, is *data-centric* (Liu & Chen, 2017; Liu et al., 2020; Zhu et al., 2021b; 2022) and training-free. The main idea is to check the agreements among multiple noisy attributes as discussed in Section 4.3.

## 7 LIMITATION AND FUTURE WORK

We point out two limitations in our work. *First*, our method shows limited improvement over directly measured metrics when auxiliary models are highly accurate and the true fairness disparity is small. *Second*, our theoretical guarantee on the superiority of our method (Corollary 2) is only theoretically proven in a simplified case. One future work is to apply the same method to unfairness mitigation algorithm in addition to evaluating fairness.

ETHICS STATEMENT

Our goal is to better study and promote fairness. Without a promising estimation method, given the increasingly stringent privacy regulations, it would be difficult for academia and industry to measure, detect, and mitigate bias in many real-world scenarios. However, we need to caution readers that, needless to say, no estimation algorithm is perfect. Theoretically, in our framework, if the transition matrix is perfectly estimated, then our method can measure fairness with 100% accuracy. However, if Assumptions 1–2 required by our estimator in Algorithm 2 do not hold, our calibrated metrics might have a non-negligible error, and therefore could be misleading. In addition, the example we use to explain terms in Theorem 1 is based on conclusions from (Angwin et al., 2016). We do not have any biased opinion on the crime rate across different racial groups. Furthermore, we are fully aware that many sensitive attributes are not binary, *e.g.* race and gender. We use the binary sensitive attributes in experiments because 1) existing works have shown that bias exists in COMPAS between race "black" and others and 2) the ground-truth gender attribute in CelebA is binary. Finally, all the data and models we use are from open-source projects, and the bias measured on them do not reflect our opinions about those projects.

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

# Appendix

The Appendix is organized as follows.

- Section A presents a summary of notations, more fairness definitions, and a clear statement of the assumption that is common in the literature. Note our framework does not rely on this assumption.
- Section B presents the full version of our theorems (for DP, EOd, EOp), corollaries, and the corresponding proofs.
- Section C shows how HOC works and analyzes why other learning-centric methods in the noisy label literature may not work in our setting.
- Section D presents more experimental results and takeaways.

The code and data for reproducing our experiments will be released after acceptance.

## A  MORE DEFINITIONS AND ASSUMPTIONS

### A.1  SUMMARY OF NOTATIONS

Table 4: Summary of key notations

| Notation | Explanation |
|---|---|
| $\mathcal{G} := \{g_1, \cdots, g_C\}$ | $C$ auxiliary models for generating noisy sensitive attributes |
| $X, Y, A$, and $\widetilde{A} := g(X)$ | Random variables of feature, label, ground-truth sensitive attribute, and noisy sensitive attributes |
| $x_n, y_n, a_n$ | The $n$-th feature, label, and ground-truth sensitive attribute in a dataset |
| $N, K, M$ | The number of instances, label classes, categories of sensitive attributes |
| $[N] := \{1, \cdots, N\}$ | A set counting from 1 to $N$ |
| $\mathcal{X}, f : \mathcal{X} \to [K]$ | Space of $X$, target model |
| $D^\circ := \{(x_n, y_n) | n \in [N]\}$ | Target dataset |
| $D := \{(x_n, y_n, a_n) | n \in [N]\}$ | $D^\circ$ with ground-truth sensitive attributes |
| $\widetilde{D} := \{(x_n, y_n, (\tilde{a}_n^1, \cdots, \tilde{a}_n^C)) | n \in [N]\}$ | $D^\circ$ with noisy sensitive attributes |
| $(X, Y, A) \sim \mathcal{D}, (X, Y, \widetilde{A}) \sim \widetilde{\mathcal{D}}$ | Distribution of $D$ and $\widetilde{D}$ |
| $u \in \{\text{DP}, \text{EOd}, \text{EOp}\}$ | A unified notation of fairness definitions, e.g., EOd, EOp, EOd |
| $\Delta^u(\mathcal{D}, f), \widetilde{\Delta}^u(\widetilde{\mathcal{D}}, f), \widehat{\Delta}^u(\widetilde{\mathcal{D}}, f)$ | True, (direct) noisy, and calibrated group fairness metrics on data distributions |
| $\Delta^u(D, f), \widetilde{\Delta}^u(\widetilde{D}, f), \widehat{\Delta}^u(\widetilde{D}, f)$ | True, (direct) noisy, and calibrated group fairness metrics on datasets |
| $\boldsymbol{H}, \boldsymbol{H}[a], \boldsymbol{H}[:, k], \boldsymbol{H}[a, k]$ | Fairness matrix, its $a$-th row, $k$-th column, $(a, k)$-th element |
| $\widetilde{\boldsymbol{H}}$ | Noisy fairness matrix with respect to $\widetilde{A}$ |
| $\boldsymbol{T}, T[a, \tilde{a}] := \mathbb{P}(\widetilde{A} = \tilde{a} | A = a)$ | Global noise transition matrix |
| $\boldsymbol{T}_k, T_k[a, \tilde{a}] := \mathbb{P}(\widetilde{A} = \tilde{a} | A = a, f(X) = k)$ | Local noise transition matrix |
| $\boldsymbol{p} := [\mathbb{P}(A = 1), \cdots, \mathbb{P}(A = M)]^\top$ | Clean prior probability |
| $\tilde{\boldsymbol{p}} := [\mathbb{P}(\widetilde{A} = 1), \cdots, \mathbb{P}(\widetilde{A} = M)]^\top$ | Clean prior probability |

### A.2  MORE FAIRNESS DEFINITIONS

We present the full version of fairness definitions and the corresponding matrix form for DP, EOd, and EOp as follows.

**Fairness Definitions.**  We consider three group fairness (Wang et al., 2020; Cotter et al., 2019) definitions and their corresponding measurable metrics: *demographic parity* (DP) (Calders et al., 2009; Chouldechova, 2017), *equalized odds* (EOd) (Woodworth et al., 2017), and *equalized opportunity* (EOp) (Hardt et al., 2016).

**Definition 1** (Demographic Parity). *The demographic parity metric of $f$ on $\mathcal{D}$ conditioned on $A$ is:*

$$\Delta^{DP}(\mathcal{D}, f) = \frac{1}{M(M-1)K} \sum_{\substack{a, a' \in [M] \\ k \in [K]}} |\mathbb{P}(f(X) = k | A = a) - \mathbb{P}(f(X) = k | A = a')|.$$

**Definition 4** (Equalized Odds). *The equalized odds metric of $f$ on $\mathcal{D}$ conditioned on $A$ is:*

$$\Delta^{EOd}(\mathcal{D}, f) = \frac{1}{M(M-1)K^2} \sum_{\substack{a, a' \in [M] \\ k \in [K], y \in [K]}} |\mathbb{P}(f(X) = k | Y = y, A = a) - \mathbb{P}(f(X) = k | Y = y, A = a')|.$$

**Definition 5** (Equalized Opportunity). *The equalized opportunity metric of $f$ on $\mathcal{D}$ conditioned on $A$ is:*

$$\Delta^{EOp}(\mathcal{D}, f) = \frac{1}{M(M-1)} \sum_{a,a' \in [M]} |\mathbb{P}(f(X) = 1|Y = 1, A = a) - \mathbb{P}(f(X) = 1|Y = 1, A = a')|.$$

**Matrix-form Metrics.** To unify three fairness metrics in a general form, we represent them with a matrix $\boldsymbol{H}$. Each column of $\boldsymbol{H}$ denotes the probability needed for evaluating fairness with respect to classifier prediction $f(X)$. For DP, $\boldsymbol{H}[:, k]$ denotes the following column vector:

$$\boldsymbol{H}[:, k] := [\mathbb{P}(f(X) = k|A = 1), \cdots, \mathbb{P}(f(X) = k|A = M)]^{\top}.$$

Similarly for EOd and EOp, let $k \otimes y := K(k-1) + y$ be the 1-d flattened index that represents the 2-d coordinate in $f(X) \times Y$, $\boldsymbol{H}[:, k \otimes y]$ is defined as the following column vector:

$$\boldsymbol{H}[:, k \otimes y] := [\mathbb{P}(f(X) = k|Y = y, A = 1), \cdots, \mathbb{P}(f(X) = k|Y = y, A = M)]^{\top}.$$

The sizes of $\boldsymbol{H}$ for DP, EOd and EOp are $M \times K$, $M \times K^2$, and $M \times 1$ respectively. The noise transition matrix related to EOd and EOp is $\boldsymbol{T}_{k \otimes y}$, where the $(a, \tilde{a})$-th element is denoted by $T_{k \otimes y}[a, \tilde{a}] := \mathbb{P}(\widetilde{A} = \tilde{a}|f(X) = k, Y = y, A = a)$.

### A.3 COMMON CONDITIONAL INDEPENDENCE ASSUMPTION IN THE LITERATURE

We present below a common conditional independence assumption in the literature (Awasthi et al., 2021; Prost et al., 2021; Fogliato et al., 2020). Note out framework successfully drops this assumption.

**Assumption 3** (Conditional Independence). *$\tilde{A}$ and $f(X)$ are conditionally independent given $A$ (and $Y$ for EOd, EOp):*

*DP:* $\mathbb{P}(\widetilde{A} = \tilde{a}|f(X) = k, A = a) = \mathbb{P}(\widetilde{A} = \tilde{a}|A = a), \forall a, \tilde{a} \in [M], k \in [K].$

$(i.e. \tilde{A} \perp\!\!\!\perp f(X)|A).$

*EOd / EOp:* $\mathbb{P}(\widetilde{A} = \tilde{a}|f(X) = k, Y = y, A = a) = \mathbb{P}(\widetilde{A} = \tilde{a}|Y = y, A = a), \forall a, \tilde{a} \in [M], k, y \in [K].$

$(i.e. \tilde{A} \perp\!\!\!\perp f(X)|Y, A).$

## B PROOFS

### B.1 FULL VERSION OF THEOREM 1 AND ITS PROOF

Denote by $\boldsymbol{T}_y$ the attribute noise transition matrix with respect to label $y$, whose $(a, \tilde{a})$-th element is $T_y[a, \tilde{a}] := \mathbb{P}(\widetilde{A} = \tilde{a}|A = a, Y = y)$. Note it is different from $\boldsymbol{T}_k$. Denote by $\boldsymbol{T}_{k \otimes y}$ the attribute noise transition matrix when $f(X) = k$ and $Y = y$, where the $(a, \tilde{a})$-th element is $\boldsymbol{T}_{k \otimes y}[a, \tilde{a}] := \mathbb{P}(\widetilde{A} = \tilde{a}|f(X) = k, Y = y, A = a)$. Denote by $\boldsymbol{p}_y := [\mathbb{P}(A = 1|Y = y), \cdots, \mathbb{P}(A = K|Y = y)]^{\top}$ and $\tilde{\boldsymbol{p}}_y := [\mathbb{P}(\widetilde{A} = 1|Y = y), \cdots, \mathbb{P}(\widetilde{A} = K|Y = y)]^{\top}$ the clean prior probabilities and noisy prior probability, respectively.

**Theorem 1** (Error Upper Bound of Noisy Metrics). *Denote by $\mathsf{Err}_u^{raw} := |\Delta^u(\widetilde{\mathcal{D}}, f) - \Delta^u(\mathcal{D}, f)|$ the estimation error of the directly measured noisy fairness metrics. Its upper bound is:*

- *DP:*

$$\mathsf{Err}_{DP}^{raw} \leq \frac{2}{K} \sum_{k \in [K]} \left( \bar{h}_k \underbrace{\|\boldsymbol{\Lambda}_{\tilde{\boldsymbol{p}}}(\boldsymbol{T}^{-1}\boldsymbol{T}_k - \boldsymbol{I})\boldsymbol{\Lambda}_{\tilde{\boldsymbol{p}}}^{-1}\|_1}_{\text{cond. indep. violation}} + \delta_k \underbrace{\|\boldsymbol{\Lambda}_{\boldsymbol{p}}\boldsymbol{T}_k\boldsymbol{\Lambda}_{\tilde{\boldsymbol{p}}}^{-1} - \boldsymbol{I}\|_1}_{\text{error of } g} \right).$$

  *where $\bar{h}_k := \frac{1}{M} \sum_{a \in [M]} H[a, k]$, $\delta_k := \max_{a \in [M]} |H[a, k] - \bar{h}_k|$.*

- EOd:

$$
\mathsf{Err}_{\mathsf{EOd}}^{\mathsf{raw}} \le \frac{2}{K^2} \sum_{k \in [K], y \in [K]} \left( \bar{h}_{k \otimes y} \underbrace{\|\boldsymbol{\Lambda}_{\tilde{\boldsymbol{p}}_y}(\boldsymbol{T}_y^{-1}\boldsymbol{T}_{k \otimes y} - \boldsymbol{I})\boldsymbol{\Lambda}_{\tilde{\boldsymbol{p}}_y}^{-1}\|_1}_{\text{cond. indep. violation}} + \delta_{k \otimes y} \underbrace{\|\boldsymbol{\Lambda}_{\boldsymbol{p}_y}\boldsymbol{T}_{k \otimes y}\boldsymbol{\Lambda}_{\tilde{\boldsymbol{p}}_y}^{-1} - \boldsymbol{I}\|_1}_{\text{error of } g} \right).
$$

where $\bar{h}_{k \otimes y} := \frac{1}{M} \sum_{a \in [M]} H[a, k \otimes y]$, $\delta_{k \otimes y} := \max_{a \in [M]} |H[a, k \otimes y] - \bar{h}_{k \otimes y}|$.

- EOp: We obtain the result for EOp by simply letting $k = 1$ and $y = 1$, i.e.,

$$
\mathsf{Err}_{\mathsf{EOp}}^{\mathsf{raw}} \le 2 \sum_{k=1, y=1} \left( \bar{h}_{k \otimes y} \underbrace{\|\boldsymbol{\Lambda}_{\tilde{\boldsymbol{p}}_y}(\boldsymbol{T}_y^{-1}\boldsymbol{T}_{k \otimes y} - \boldsymbol{I})\boldsymbol{\Lambda}_{\tilde{\boldsymbol{p}}_y}^{-1}\|_1}_{\text{cond. indep. violation}} + \delta_{k \otimes y} \underbrace{\|\boldsymbol{\Lambda}_{\boldsymbol{p}_y}\boldsymbol{T}_{k \otimes y}\boldsymbol{\Lambda}_{\tilde{\boldsymbol{p}}_y}^{-1} - \boldsymbol{I}\|_1}_{\text{error of } g} \right).
$$

where $\bar{h}_{k \otimes y} := \frac{1}{M} \sum_{a \in [M]} H[a, k \otimes y]$, $\delta_{k \otimes y} := \max_{a \in [M]} |H[a, k \otimes y] - \bar{h}_{k \otimes y}|$.

*Proof.* The following proof builds on the relationship derived in the proof for Theorem 2. We encourage readers to check Appendix B.2 before the following proof.

Recall $\boldsymbol{T}_y[a, a'] := \mathbb{P}(\widetilde{A} = a'|A = a, Y = y)$. Note

$$
\boldsymbol{\Lambda}_{\tilde{\boldsymbol{p}}_y}\boldsymbol{1} = \boldsymbol{T}_y^\top \boldsymbol{\Lambda}_{\boldsymbol{p}_y}\boldsymbol{1} \Leftrightarrow (\boldsymbol{T}_y^\top)^{-1}\boldsymbol{\Lambda}_{\tilde{\boldsymbol{p}}_y}\boldsymbol{1} = \boldsymbol{\Lambda}_{\boldsymbol{p}_y}\boldsymbol{1}.
$$

Denote by

$$
\boldsymbol{H}[:, k \otimes y] = \bar{h}_{k \otimes y}\boldsymbol{1} + \boldsymbol{v}_{k \otimes y},
$$

where $\bar{h}_{k \otimes y} := \frac{1}{M} \sum_{a \in [M]} \mathbb{P}(f(X) = k|A = a, Y = y)$. We have

$$
\boldsymbol{\Lambda}_{\boldsymbol{p}_y}\boldsymbol{H}[:, k \otimes y] = \bar{h}_{k \otimes y}\boldsymbol{\Lambda}_{\boldsymbol{p}_y}\boldsymbol{1} + \boldsymbol{\Lambda}_{\boldsymbol{p}_y}\boldsymbol{v}_{k \otimes y} = \bar{h}_{k \otimes y}(\boldsymbol{T}_y^\top)^{-1}\boldsymbol{\Lambda}_{\tilde{\boldsymbol{p}}_y}\boldsymbol{1} + \boldsymbol{\Lambda}_{\boldsymbol{p}_y}\boldsymbol{v}_{k \otimes y}.
$$

We further have

$$
\begin{aligned}
&\widetilde{\boldsymbol{H}}[: k \otimes y] \\
&= \left(\boldsymbol{\Lambda}_{\tilde{\boldsymbol{p}}_y}^{-1}\boldsymbol{T}_{k \otimes y}^\top \boldsymbol{\Lambda}_{\boldsymbol{p}_y} - \boldsymbol{I}\right)\boldsymbol{H}[:, k \otimes y] + \boldsymbol{H}[:, k \otimes y] \\
&= \bar{h}_{k \otimes y}\boldsymbol{\Lambda}_{\tilde{\boldsymbol{p}}_y}^{-1}\boldsymbol{T}_{k \otimes y}^\top(\boldsymbol{T}_y^\top)^{-1}\boldsymbol{\Lambda}_{\tilde{\boldsymbol{p}}_y}\boldsymbol{1} + \boldsymbol{\Lambda}_{\tilde{\boldsymbol{p}}_y}^{-1}\boldsymbol{T}_{k \otimes y}^\top \boldsymbol{\Lambda}_{\boldsymbol{p}_y}\boldsymbol{v}_{k \otimes y} - \bar{h}_{k \otimes y}\boldsymbol{1} - \boldsymbol{v}_{k \otimes y} + \boldsymbol{H}[:, k \otimes y] \\
&= \bar{h}_{k \otimes y}\boldsymbol{\Lambda}_{\tilde{\boldsymbol{p}}_y}^{-1}\left(\boldsymbol{T}_{k \otimes y}^\top(\boldsymbol{T}_y^\top)^{-1} - \boldsymbol{I}\right)\boldsymbol{\Lambda}_{\tilde{\boldsymbol{p}}_y}\boldsymbol{1} + \left(\boldsymbol{\Lambda}_{\tilde{\boldsymbol{p}}_y}^{-1}\boldsymbol{T}_{k \otimes y}^\top \boldsymbol{\Lambda}_{\boldsymbol{p}_y} - \boldsymbol{I}\right)\boldsymbol{v}_{k \otimes y} + \boldsymbol{H}[:, k \otimes y].
\end{aligned}
$$

Noting $|A| - |B| \le |A + B| \le |A| + |B|$, we have $||A + B| - |B|| \le |A|$. Therefore,

$$
\begin{aligned}
&\left| \left|(\boldsymbol{e}_{\tilde{a}} - \boldsymbol{e}_{\tilde{a}'})^\top \widetilde{\boldsymbol{H}}[: k \otimes y]\right| - \left|(\boldsymbol{e}_{\tilde{a}} - \boldsymbol{e}_{\tilde{a}'})^\top \boldsymbol{H}[: k \otimes y]\right| \right| \\
&\le \bar{h}_{k \otimes y}\left|(\boldsymbol{e}_{\tilde{a}} - \boldsymbol{e}_{\tilde{a}'})^\top \boldsymbol{\Lambda}_{\tilde{\boldsymbol{p}}_y}^{-1}\left(\boldsymbol{T}_y^{-1}\boldsymbol{T}_{k \otimes y} - \boldsymbol{I}\right)^\top \boldsymbol{\Lambda}_{\tilde{\boldsymbol{p}}_y}\boldsymbol{1}\right| \qquad \text{(Term 1)} \\
&\quad + \left|(\boldsymbol{e}_{\tilde{a}} - \boldsymbol{e}_{\tilde{a}'})^\top \left(\boldsymbol{\Lambda}_{\tilde{\boldsymbol{p}}_y}^{-1}\boldsymbol{T}_{k \otimes y}^\top \boldsymbol{\Lambda}_{\boldsymbol{p}_y} - \boldsymbol{I}\right)\boldsymbol{v}_{k \otimes y}\right|. \qquad \text{(Term 2)}
\end{aligned}
$$

Term-1 and Term-2 can be upper bounded as follows.

**Term 1:** With the Hölder's inequality, we have

$$
\begin{aligned}
&\bar{h}_{k \otimes y}\left|(\boldsymbol{e}_{\tilde{a}} - \boldsymbol{e}_{\tilde{a}'})^\top \boldsymbol{\Lambda}_{\tilde{\boldsymbol{p}}_y}^{-1}\left(\boldsymbol{T}_y^{-1}\boldsymbol{T}_{k \otimes y} - \boldsymbol{I}\right)^\top \boldsymbol{\Lambda}_{\tilde{\boldsymbol{p}}_y}\boldsymbol{1}\right| \\
&\le \bar{h}_{k \otimes y}\|\boldsymbol{e}_{\tilde{a}} - \boldsymbol{e}_{\tilde{a}'}\|_1\left\|\boldsymbol{\Lambda}_{\tilde{\boldsymbol{p}}_y}^{-1}\left(\boldsymbol{T}_y^{-1}\boldsymbol{T}_{k \otimes y} - \boldsymbol{I}\right)^\top \boldsymbol{\Lambda}_{\tilde{\boldsymbol{p}}_y}\boldsymbol{1}\right\|_\infty \\
&\le 2\bar{h}_{k \otimes y}\left\|\boldsymbol{\Lambda}_{\tilde{\boldsymbol{p}}_y}^{-1}\left(\boldsymbol{T}_y^{-1}\boldsymbol{T}_{k \otimes y} - \boldsymbol{I}\right)^\top \boldsymbol{\Lambda}_{\tilde{\boldsymbol{p}}_y}\boldsymbol{1}\right\|_\infty \\
&\le 2\bar{h}_{k \otimes y}\left\|\boldsymbol{\Lambda}_{\tilde{\boldsymbol{p}}_y}^{-1}\left(\boldsymbol{T}_y^{-1}\boldsymbol{T}_{k \otimes y} - \boldsymbol{I}\right)^\top \boldsymbol{\Lambda}_{\tilde{\boldsymbol{p}}_y}\right\|_\infty \\
&= 2\bar{h}_{k \otimes y}\left\|\boldsymbol{\Lambda}_{\tilde{\boldsymbol{p}}_y}\left(\boldsymbol{T}_y^{-1}\boldsymbol{T}_{k \otimes y} - \boldsymbol{I}\right)\boldsymbol{\Lambda}_{\tilde{\boldsymbol{p}}_y}^{-1}\right\|_1
\end{aligned}
$$

**Term 2:** Denote by $\delta_{k\otimes y} := \max\limits_{a\in[M]} |H[a, k\otimes y] - \bar{h}_{k\otimes y}|$, which is the largest absolute offset from its mean. With the Hölder's inequality, we have

$$\left| (\boldsymbol{e}_{\tilde{a}} - \boldsymbol{e}_{\tilde{a}'})^\top \left( \boldsymbol{\Lambda}_{\tilde{\boldsymbol{p}}_y}^{-1} \boldsymbol{T}_{k\otimes y}^\top \boldsymbol{\Lambda}_{\boldsymbol{p}_y} - \boldsymbol{I} \right) \boldsymbol{v}_{k\otimes y} \right|$$

$$\leq \|\boldsymbol{e}_{\tilde{a}} - \boldsymbol{e}_{\tilde{a}'}\|_1 \left\| \left( \boldsymbol{\Lambda}_{\tilde{\boldsymbol{p}}_y}^{-1} \boldsymbol{T}_{k\otimes y}^\top \boldsymbol{\Lambda}_{\boldsymbol{p}_y} - \boldsymbol{I} \right) \boldsymbol{v}_{k\otimes y} \right\|_\infty$$

$$\leq 2 \left\| \left( \boldsymbol{\Lambda}_{\tilde{\boldsymbol{p}}_y}^{-1} \boldsymbol{T}_{k\otimes y}^\top \boldsymbol{\Lambda}_{\boldsymbol{p}_y} - \boldsymbol{I} \right) \boldsymbol{v}_{k\otimes y} \right\|_\infty$$

$$\leq 2\delta_{k\otimes y} \left\| \boldsymbol{\Lambda}_{\tilde{\boldsymbol{p}}_y}^{-1} \boldsymbol{T}_{k\otimes y}^\top \boldsymbol{\Lambda}_{\boldsymbol{p}_y} - \boldsymbol{I} \right\|_\infty$$

$$= 2\delta_{k\otimes y} \left\| \boldsymbol{\Lambda}_{\boldsymbol{p}_y} \boldsymbol{T}_{k\otimes y} \boldsymbol{\Lambda}_{\tilde{\boldsymbol{p}}_y}^{-1} - \boldsymbol{I} \right\|_1$$

**Wrap-up:**

$$\left| \left| (\boldsymbol{e}_{\tilde{a}} - \boldsymbol{e}_{\tilde{a}'})^\top \widetilde{\boldsymbol{H}}[: k \otimes y] \right| - \left| (\boldsymbol{e}_{\tilde{a}} - \boldsymbol{e}_{\tilde{a}'})^\top \boldsymbol{H}[: k \otimes y] \right| \right|$$

$$\leq 2\bar{h}_{k\otimes y} \left\| \boldsymbol{\Lambda}_{\tilde{\boldsymbol{p}}_y} \left( \boldsymbol{T}_y^{-1} \boldsymbol{T}_{k\otimes y} - \boldsymbol{I} \right) \boldsymbol{\Lambda}_{\tilde{\boldsymbol{p}}_y}^{-1} \right\|_1 + 2\delta_{k\otimes y} \left\| \boldsymbol{\Lambda}_{\boldsymbol{p}_y} \boldsymbol{T}_{k\otimes y} \boldsymbol{\Lambda}_{\tilde{\boldsymbol{p}}_y}^{-1} - \boldsymbol{I} \right\|_1.$$

Denote by $\widetilde{\Delta}_{k\otimes y}^{\tilde{a},\tilde{a}'} := |\widetilde{\boldsymbol{H}}[\tilde{a}, k \otimes y] - \widetilde{\boldsymbol{H}}[\tilde{a}', k \otimes y]|$ the noisy disparity and $\Delta_{k\otimes y}^{\tilde{a},\tilde{a}'} := |\boldsymbol{H}[\tilde{a}, k \otimes y] - \boldsymbol{H}[\tilde{a}', k \otimes y]|$ the clean disparity between attributes $\tilde{a}$ and $\tilde{a}'$ in the case when $f(X) = k$ and $Y = y$. We have

$$\left| \widetilde{\Delta}^{\mathsf{EOd}}(\widetilde{\mathcal{D}}, f) - \Delta^{\mathsf{EOd}}(\mathcal{D}, f) \right|$$

$$\leq \frac{1}{M(M-1)K^2} \sum_{\tilde{a},\tilde{a}'\in[M],k,y\in[K]} \left| \widetilde{\Delta}_{k\otimes y}^{\tilde{a},\tilde{a}'} - \Delta_{k\otimes y}^{\tilde{a},\tilde{a}'} \right|$$

$$\leq \frac{2}{M(M-1)K^2} \sum_{\tilde{a},\tilde{a}'\in[M],k,y\in[K]} \left( \bar{h}_{k\otimes y} \left\| \boldsymbol{\Lambda}_{\tilde{\boldsymbol{p}}_y} \left( \boldsymbol{T}_y^{-1} \boldsymbol{T}_{k\otimes y} - \boldsymbol{I} \right) \boldsymbol{\Lambda}_{\tilde{\boldsymbol{p}}_y}^{-1} \right\|_1 + \delta_{k\otimes y} \left\| \boldsymbol{\Lambda}_{\boldsymbol{p}_y} \boldsymbol{T}_{k\otimes y} \boldsymbol{\Lambda}_{\tilde{\boldsymbol{p}}_y}^{-1} - \boldsymbol{I} \right\|_1 \right)$$

$$= \frac{2}{K^2} \sum_{k,y\in[K]} \left( \bar{h}_{k\otimes y} \left\| \boldsymbol{\Lambda}_{\tilde{\boldsymbol{p}}_y} \left( \boldsymbol{T}_y^{-1} \boldsymbol{T}_{k\otimes y} - \boldsymbol{I} \right) \boldsymbol{\Lambda}_{\tilde{\boldsymbol{p}}_y}^{-1} \right\|_1 + \delta_{k\otimes y} \left\| \boldsymbol{\Lambda}_{\boldsymbol{p}_y} \boldsymbol{T}_{k\otimes y} \boldsymbol{\Lambda}_{\tilde{\boldsymbol{p}}_y}^{-1} - \boldsymbol{I} \right\|_1 \right).$$

The results for $\mathsf{DP}$ can be obtained by dropping the dependence on $Y = y$, and the results for $\mathsf{EOp}$ can be obtained by letting $k = 1$ and $y = 1$. $\qquad\square$

### B.2 FULL VERSION OF THEOREM 2 AND ITS PROOF

Recall $\boldsymbol{p}$, $\tilde{\boldsymbol{p}}$, $\boldsymbol{T}$ and $\boldsymbol{T}_k$ are clean prior, noisy prior, global transition matrix, and local transition matrix defined in Sec. 2. Denote by $\boldsymbol{\Lambda}_{\tilde{\boldsymbol{p}}}$ and $\boldsymbol{\Lambda}_{\boldsymbol{p}}$ the square diagonal matrices constructed from $\tilde{\boldsymbol{p}}$ and $\boldsymbol{p}$.

**Theorem 2.** *[Closed-form relationship ($\mathsf{DP},\mathsf{EOd},\mathsf{EOp}$)] The relationship between the true fairness vector $\boldsymbol{h}^u$ and the corresponding noisy fairness vector $\tilde{\boldsymbol{h}}^u$ writes as*

$$\boldsymbol{h}^u = (\boldsymbol{T}^{u\top} \boldsymbol{\Lambda}_{\boldsymbol{p}^u})^{-1} \boldsymbol{\Lambda}_{\tilde{\boldsymbol{p}}^u} \tilde{\boldsymbol{h}}^u, \quad \forall u \in \{\mathsf{DP}, \mathsf{EOd}, \mathsf{EOp}\},$$

*where $\boldsymbol{\Lambda}_{\tilde{\boldsymbol{p}}^u}$ and $\boldsymbol{\Lambda}_{\boldsymbol{p}^u}$ denote the square diagonal matrix constructed from $\tilde{\boldsymbol{p}}^u$ and $\boldsymbol{p}^u$, $u$ unifies different fairness metrics. Particularly,*

- *$\mathsf{DP}$ ($\forall k \in [K]$): $\boldsymbol{p}^{\mathsf{DP}} := [\mathbb{P}(A = 1), \cdots, \mathbb{P}(A = M)]^\top$, $\tilde{\boldsymbol{p}}^{\mathsf{DP}} := [\mathbb{P}(\widetilde{A} = 1), \cdots, \mathbb{P}(\widetilde{A} = M)]^\top$. $\boldsymbol{T}^{\mathsf{DP}} := \boldsymbol{T}_k$, where the $(a, \tilde{a})$-th element of $\boldsymbol{T}_k$ is $T_k[a, \tilde{a}] := \mathbb{P}(\widetilde{A} = \tilde{a} | f(X) = k, A = a)$.*

$$\boldsymbol{h}^{\mathsf{DP}} := \boldsymbol{H}[:, k] := [\mathbb{P}(f(X) = k | A = 1), \cdots, \mathbb{P}(f(X) = k | A = M)]^\top$$

$$\tilde{\boldsymbol{h}}^{\mathsf{DP}} := \widetilde{\boldsymbol{H}}[:, k] := [\mathbb{P}(f(X) = k | \widetilde{A} = 1), \cdots, \mathbb{P}(f(X) = k | \widetilde{A} = M)]^\top.$$

- *EOd and EOp* ($\forall k, y \in [K], u \in \{$*EOd*, *EOp*$\}$)*:* $\forall k, y \in [K]$*:* $k \otimes y := K(k-1) + y$, $\boldsymbol{p}^u := \boldsymbol{p}_y := [\mathbb{P}(A = 1|Y = y), \cdots, \mathbb{P}(A = M|Y = y)]^\top$, $\tilde{\boldsymbol{p}}^u := \tilde{\boldsymbol{p}}_y := [\mathbb{P}(\widetilde{A} = 1|Y = y), \cdots, \mathbb{P}(\widetilde{A} = M|Y = y)]^\top$. $\boldsymbol{T}^u := \boldsymbol{T}_{k \otimes y}$, *where the* $(a, \tilde{a})$*-th element of* $\boldsymbol{T}_{k \otimes y}$ *is* $T_{k \otimes y}[a, \tilde{a}] := \mathbb{P}(\widetilde{A} = \tilde{a}|f(X) = k, Y = y, A = a)$.

$$\boldsymbol{h}^u := \boldsymbol{H}[:, k \otimes y] := [\mathbb{P}(f(X) = k|Y = y, A = 1), \cdots, \mathbb{P}(f(X) = k|Y = y, A = M)]^\top$$
$$\tilde{\boldsymbol{h}}^u := \widetilde{\boldsymbol{H}}[:, k \otimes y] := [\mathbb{P}(f(X) = k|Y = y, \widetilde{A} = 1), \cdots, \mathbb{P}(f(X) = k|Y = y, \widetilde{A} = M)]^\top.$$

*Proof.* We first prove the theorem for DP, then for EOd and EOp.

**Proof for DP.** In DP, each element of $\tilde{\boldsymbol{h}}^{\mathsf{DP}}$ satisfies:

$$\mathbb{P}(f(X) = k|\widetilde{A} = \tilde{a})$$
$$= \frac{\sum_{a \in [M]} \mathbb{P}(f(X) = k, \widetilde{A} = \tilde{a}, A = a)}{\mathbb{P}(\widetilde{A} = \tilde{a})}$$
$$= \frac{\sum_{a \in [M]} \mathbb{P}(\widetilde{A} = \tilde{a}|f(X) = k, A = a) \cdot \mathbb{P}(A = a) \cdot \mathbb{P}(f(X) = k|A = a)}{\mathbb{P}(\widetilde{A} = \tilde{a})}$$

Recall $\boldsymbol{T}_k$ is the attribute noise transition matrix when $f(X) = k$, where the $(a, \tilde{a})$-th element is $T_k[a, \tilde{a}] := \mathbb{P}(\widetilde{A} = \tilde{a}|f(X) = k, A = a)$. Recall $\boldsymbol{p} := [\mathbb{P}(A = 1), \cdots, \mathbb{P}(A = M)]^\top$ and $\tilde{\boldsymbol{p}} := [\mathbb{P}(\widetilde{A} = 1), \cdots, \mathbb{P}(\widetilde{A} = M)]^\top$ the clean prior probabilities and noisy prior probability, respectively. The above equation can be re-written as a matrix form as

$$\widetilde{\boldsymbol{H}}[:, k] = \boldsymbol{\Lambda}_{\tilde{\boldsymbol{p}}}^{-1} \boldsymbol{T}_k^\top \boldsymbol{\Lambda}_{\boldsymbol{p}} \boldsymbol{H}[:, k],$$

which is equivalent to

$$\boldsymbol{H}[:, k] = ((\boldsymbol{T}_k^\top) \boldsymbol{\Lambda}_{\boldsymbol{p}})^{-1} \boldsymbol{\Lambda}_{\tilde{\boldsymbol{p}}} \widetilde{\boldsymbol{H}}[:, k].$$

**Proof for EOd, EOp.** In EOd or EOp, each element of $\tilde{\boldsymbol{h}}^u$ satisfies:

$$\mathbb{P}(f(X) = k|Y = y, \widetilde{A} = \tilde{a})$$
$$= \frac{\mathbb{P}(f(X) = k, Y = y, \widetilde{A} = \tilde{a})}{\mathbb{P}(Y = y, \widetilde{A} = \tilde{a})}$$
$$= \frac{\sum_{a \in [M]} \mathbb{P}(f(X) = k, Y = y, \widetilde{A} = \tilde{a}, A = a)}{\mathbb{P}(Y = y, \widetilde{A} = \tilde{a})}$$
$$= \frac{\sum_{a \in [M]} \mathbb{P}(\widetilde{A} = \tilde{a}|f(X) = k, Y = y, A = a) \cdot \mathbb{P}(Y = y, A = a) \cdot \mathbb{P}(f(X) = k|Y = y, A = a)}{\mathbb{P}(Y = y, \widetilde{A} = \tilde{a})}$$

Denote by $\boldsymbol{T}_{k \otimes y}$ the attribute noise transition matrix when $f(X) = k$ and $Y = y$, where the $(a, \tilde{a})$-th element is $\boldsymbol{T}_{k \otimes y}[a, \tilde{a}] := \mathbb{P}(\widetilde{A} = \tilde{a}|f(X) = k, Y = y, A = a)$. Denote by $\boldsymbol{p}_y := [\mathbb{P}(A = 1|Y = y), \cdots, \mathbb{P}(A = K|Y = y)]^\top$ and $\tilde{\boldsymbol{p}}_y := [\mathbb{P}(\widetilde{A} = 1|Y = y), \cdots, \mathbb{P}(\widetilde{A} = K|Y = y)]^\top$ the clean prior probabilities and noisy prior probability, respectively. The above equation can be re-written as a matrix form as

$$\widetilde{\boldsymbol{H}}[:, k] = \boldsymbol{\Lambda}_{\tilde{\boldsymbol{p}}_y}^{-1} \boldsymbol{T}_{k \otimes y}^\top \boldsymbol{\Lambda}_{\boldsymbol{p}_y} \boldsymbol{H}[:, k],$$

which is equivalent to

$$\boldsymbol{H}[:, k] = (\boldsymbol{T}_{k \otimes y}^\top \boldsymbol{\Lambda}_{\boldsymbol{p}_y})^{-1} \boldsymbol{\Lambda}_{\tilde{\boldsymbol{p}}_y} \widetilde{\boldsymbol{H}}[:, k].$$

**Wrap-up.** We can conclude the proof by unifying the above two results with $u$. $\qquad\square$

## B.3 PROOF FOR COROLLARY 1

*Proof.* When the conditional independence (Assumption 3)

$$\mathbb{P}(\widetilde{A} = a'|A = a, Y = y) = \mathbb{P}(\widetilde{A} = a'|A = a, f(X) = k, Y = y), \forall a', a \in [M]$$

holds, we have $\boldsymbol{T}_y = \boldsymbol{T}_{k\otimes y}$ and Term-1 in Theorem 1 can be dropped. For Term-2, to get a tight bound in this specific case, we apply the Hölder's inequality by using $l_\infty$ norm on $\boldsymbol{e}_{\tilde{a}} - \boldsymbol{e}_{\tilde{a}'}$, i.e.,

$$\left| (\boldsymbol{e}_{\tilde{a}} - \boldsymbol{e}_{\tilde{a}'})^\top \left( \boldsymbol{\Lambda}_{\tilde{\boldsymbol{p}}_y}^{-1} \boldsymbol{T}_{k\otimes y}^\top \boldsymbol{\Lambda}_{\boldsymbol{p}_y} - \boldsymbol{I} \right) \boldsymbol{v}_{k\otimes y} \right|$$

$$\leq \|\boldsymbol{e}_{\tilde{a}} - \boldsymbol{e}_{\tilde{a}'}\|_\infty \left\| \left( \boldsymbol{\Lambda}_{\tilde{\boldsymbol{p}}_y}^{-1} \boldsymbol{T}_{k\otimes y}^\top \boldsymbol{\Lambda}_{\boldsymbol{p}_y} - \boldsymbol{I} \right) \boldsymbol{v}_{k\otimes y} \right\|_1$$

$$= \left\| \left( \boldsymbol{\Lambda}_{\tilde{\boldsymbol{p}}_y}^{-1} \boldsymbol{T}_{k\otimes y}^\top \boldsymbol{\Lambda}_{\boldsymbol{p}_y} - \boldsymbol{I} \right) \boldsymbol{v}_{k\otimes y} \right\|_1$$

$$\leq K \cdot \delta_{k\otimes y} \left\| \boldsymbol{\Lambda}_{\tilde{\boldsymbol{p}}_y}^{-1} \boldsymbol{T}_{k\otimes y}^\top \boldsymbol{\Lambda}_{\boldsymbol{p}_y} - \boldsymbol{I} \right\|_1$$

$$= K \cdot \delta_{k\otimes y} \left\| \boldsymbol{\Lambda}_{\boldsymbol{p}_y} \boldsymbol{T}_{k\otimes y} \boldsymbol{\Lambda}_{\tilde{\boldsymbol{p}}_y}^{-1} - \boldsymbol{I} \right\|_\infty$$

Therefore,

$$\left| \widetilde{\Delta}^{\mathsf{EOd}}(\widetilde{\mathcal{D}}, f) - \Delta^{\mathsf{EOd}}(\mathcal{D}, f) \right|$$

$$\leq \frac{1}{K} \sum_{k,y\in[K]} \delta_{k\otimes y} \left\| \boldsymbol{\Lambda}_{\boldsymbol{p}_y} \boldsymbol{T}_{k\otimes y} \boldsymbol{\Lambda}_{\tilde{\boldsymbol{p}}_y}^{-1} - \boldsymbol{I} \right\|_\infty$$

$$= \frac{1}{K} \sum_{k,y\in[K]} \delta_{k\otimes y} \left\| \boldsymbol{\Lambda}_{\boldsymbol{p}_y} \boldsymbol{T}_y \boldsymbol{\Lambda}_{\tilde{\boldsymbol{p}}_y}^{-1} - \boldsymbol{I} \right\|_\infty$$

$$= \frac{1}{K} \sum_{k,y\in[K]} \delta_{k\otimes y} \left\| \check{\boldsymbol{T}}_y - \boldsymbol{I} \right\|_\infty,$$

where $\check{T}_y[a, \tilde{a}] = \mathbb{P}(A = a|\widetilde{A} = \tilde{a}, Y = y)$.

**Special binary case in DP** In addition to the conditional independence, when the sensitive attribute is binary and the label class is binary, considering DP, we have

$$\left| \widetilde{\Delta}^{\mathsf{DP}}(\widetilde{\mathcal{D}}, f) - \Delta^{\mathsf{DP}}(\mathcal{D}, f) \right| \leq 2\delta_k \left\| \check{\boldsymbol{T}} - \boldsymbol{I} \right\|_\infty,$$

where $\check{T}_y[a, \tilde{a}] = \mathbb{P}(A = a|\widetilde{A} = \tilde{a})$. Let $\check{T}_y[1, 2] = e_1, \check{T}_y[2, 1] = e_2$, we know

$$\check{\boldsymbol{T}} := \begin{pmatrix} 1 - e_2 & e_1 \\ e_2 & 1 - e_1 \end{pmatrix}$$

and

$$\left| \widetilde{\Delta}^{\mathsf{DP}}(\widetilde{\mathcal{D}}, f) - \Delta^{\mathsf{DP}}(\mathcal{D}, f) \right| \leq 2\delta_k \cdot (e_1 + e_2).$$

Note the equality in above inequality always holds. To prove it, firstly we note

$$\mathbb{P}(f(X) = k|\widetilde{A} = \tilde{a})$$

$$= \frac{\sum_{a\in[M]} \mathbb{P}(f(X) = k, \widetilde{A} = \tilde{a}, A = a)}{\mathbb{P}(\widetilde{A} = \tilde{a})}$$

$$= \frac{\sum_{a\in[M]} \mathbb{P}(\widetilde{A} = \tilde{a}|f(X) = k, A = a) \cdot \mathbb{P}(A = a) \cdot \mathbb{P}(f(X) = k|A = a)}{\mathbb{P}(\widetilde{A} = \tilde{a})}$$

$$= \frac{\sum_{a\in[M]} \mathbb{P}(\widetilde{A} = \tilde{a}|A = a) \cdot \mathbb{P}(A = a) \cdot \mathbb{P}(f(X) = k|A = a)}{\mathbb{P}(\widetilde{A} = \tilde{a})}$$

$$= \sum_{a\in[M]} \mathbb{P}(A = a|\widetilde{A} = \tilde{a}) \cdot \mathbb{P}(f(X) = k|A = a),$$

*i.e.* $\widetilde{\boldsymbol{H}}[:, k] = \check{\boldsymbol{T}}^{\top} \boldsymbol{H}[:, k]$. Denote by $\boldsymbol{H}[:, 1] = [h, h']^{\top}$. We have ($\tilde{a} \neq \tilde{a}'$)

$$\left| (\boldsymbol{e}_{\tilde{a}} - \boldsymbol{e}_{\tilde{a}'})^{\top} \widetilde{\boldsymbol{H}}[:, 1] \right| = |h - h'| \cdot |1 - e_1 - e_2|,$$

and

$$\left| (\boldsymbol{e}_{\tilde{a}} - \boldsymbol{e}_{\tilde{a}'})^{\top} \boldsymbol{H}[:, 1] \right| = |h - h'|.$$

Therefore, letting $\tilde{a} = 1, \tilde{a} = 2$, we have

$$\left| \widetilde{\Delta}^{\mathsf{DP}}(\widetilde{\mathcal{D}}, f) - \Delta^{\mathsf{DP}}(\mathcal{D}, f) \right|$$

$$= \frac{1}{2} \sum_{k \in \{1,2\}} \left| \left| (\boldsymbol{e}_1 - \boldsymbol{e}_2)^{\top} \widetilde{\boldsymbol{H}}[:, k] \right| - \left| (\boldsymbol{e}_1 - \boldsymbol{e}_2)^{\top} \boldsymbol{H}[:, k] \right| \right|$$

$$= \left| \left| (\boldsymbol{e}_1 - \boldsymbol{e}_2)^{\top} \widetilde{\boldsymbol{H}}[:, 1] \right| - \left| (\boldsymbol{e}_1 - \boldsymbol{e}_2)^{\top} \boldsymbol{H}[:, 1] \right| \right|$$

$$= |h - h'| \cdot |e_1 + e_2|$$

$$= 2\delta \cdot (e_1 + e_2),$$

where $\delta = |\mathbb{P}(f(X) = 1 | A = 1) - \mathbb{P}(f(X) = 1 | A = 2)| / 2$. Therefore, the equality holds.

$\square$

## B.4 PROOF FOR THEOREM 3

**Theorem 3** (Error upper bound of calibrated metrics). *Denote the error of the calibrated fairness metrics by* $\mathsf{Err}_u^{cal} := |\widehat{\Delta}^u(\widetilde{\mathcal{D}}, f) - \Delta^u(\mathcal{D}, f)|$. *It can be upper bounded as:*

- *DP:*

$$\mathsf{Err}_{DP}^{cal} \leq \frac{2}{K} \sum_{k \in [K]} \left\| \boldsymbol{\Lambda}_{\boldsymbol{p}}^{-1} \right\|_1 \left\| \boldsymbol{\Lambda}_{\boldsymbol{p}} \boldsymbol{H}[:, k] \right\|_{\infty} \varepsilon(\widehat{\boldsymbol{T}}_k, \hat{\boldsymbol{p}}),$$

*where* $\varepsilon(\widehat{\boldsymbol{T}}_k, \hat{\boldsymbol{p}}) := \|\boldsymbol{\Lambda}_{\hat{\boldsymbol{p}}}^{-1} \boldsymbol{\Lambda}_{\boldsymbol{p}} - \boldsymbol{I}\|_1 \|\boldsymbol{T}_k \widehat{\boldsymbol{T}}_k^{-1}\|_1 + \|\boldsymbol{I} - \boldsymbol{T}_k \widehat{\boldsymbol{T}}_k^{-1}\|_1$ *is the error induced by calibration.*

- *EOd:*

$$\mathsf{Err}_{EOd}^{cal} \leq \frac{2}{K^2} \sum_{k \in [K], y \in [K]} \left\| \boldsymbol{\Lambda}_{\boldsymbol{p}_y}^{-1} \right\|_1 \left\| \boldsymbol{\Lambda}_{\boldsymbol{p}_y} \boldsymbol{H}[:, k \otimes y] \right\|_{\infty} \varepsilon(\widehat{\boldsymbol{T}}_{k \otimes y}, \hat{\boldsymbol{p}}_y),$$

*where* $\varepsilon(\widehat{\boldsymbol{T}}_{k \otimes y}, \hat{\boldsymbol{p}}_y) := \|\boldsymbol{\Lambda}_{\hat{\boldsymbol{p}}_y}^{-1} \boldsymbol{\Lambda}_{\boldsymbol{p}_y} - \boldsymbol{I}\|_1 \|\boldsymbol{T}_{k \otimes y} \widehat{\boldsymbol{T}}_{k \otimes y}^{-1}\|_1 + \|\boldsymbol{I} - \boldsymbol{T}_{k \otimes y} \widehat{\boldsymbol{T}}_{k \otimes y}^{-1}\|_1$ *is the error induced by calibration.*

- *EOp:*

$$\mathsf{Err}_{EOp}^{cal} \leq 2 \sum_{k=1, y=1} \left\| \boldsymbol{\Lambda}_{\boldsymbol{p}_y}^{-1} \right\|_1 \left\| \boldsymbol{\Lambda}_{\boldsymbol{p}_y} \boldsymbol{H}[:, k \otimes y] \right\|_{\infty} \varepsilon(\widehat{\boldsymbol{T}}_{k \otimes y}, \hat{\boldsymbol{p}}_y),$$

*where* $\varepsilon(\widehat{\boldsymbol{T}}_{k \otimes y}, \hat{\boldsymbol{p}}_y) := \|\boldsymbol{\Lambda}_{\hat{\boldsymbol{p}}_y}^{-1} \boldsymbol{\Lambda}_{\boldsymbol{p}_y} - \boldsymbol{I}\|_1 \|\boldsymbol{T}_{k \otimes y} \widehat{\boldsymbol{T}}_{k \otimes y}^{-1}\|_1 + \|\boldsymbol{I} - \boldsymbol{T}_{k \otimes y} \widehat{\boldsymbol{T}}_{k \otimes y}^{-1}\|_1$ *is the error induced by calibration.*

*Proof.* We prove with $\mathsf{EOd}$.

Consider the case when $f(X) = k$ and $Y = y$. For ease of notations, we use $\widehat{\boldsymbol{T}}$ to denote the estimated local transition matrix (should be $\widehat{\boldsymbol{T}}_{k \otimes y}$). Denote the noisy (clean) fairness vectors with respect to $f(X) = k$ and $Y = y$ by $\tilde{\boldsymbol{h}}$ ($\boldsymbol{h}$). The error can be decomposed by

$$\left| \left| (\boldsymbol{e}_a - \boldsymbol{e}_{a'})^{\top} \left( \boldsymbol{\Lambda}_{\hat{\boldsymbol{p}}_y}^{-1} (\widehat{\boldsymbol{T}}^{\top})^{-1} \boldsymbol{\Lambda}_{\tilde{\boldsymbol{p}}_y} \tilde{\boldsymbol{h}} \right) \right| - \left| (\boldsymbol{e}_a - \boldsymbol{e}_{a'})^{\top} \left( \boldsymbol{\Lambda}_{\boldsymbol{p}_y}^{-1} (\boldsymbol{T}_{k \otimes y}^{\top})^{-1} \boldsymbol{\Lambda}_{\tilde{\boldsymbol{p}}_y} \tilde{\boldsymbol{h}} \right) \right| \right|$$

$$= \underbrace{\left| (\boldsymbol{e}_a - \boldsymbol{e}_{a'})^{\top} \left( (\boldsymbol{\Lambda}_{\hat{\boldsymbol{p}}_y}^{-1} - \boldsymbol{\Lambda}_{\boldsymbol{p}_y}^{-1}) (\widehat{\boldsymbol{T}}^{\top})^{-1} \boldsymbol{\Lambda}_{\tilde{\boldsymbol{p}}_y} \tilde{\boldsymbol{h}} \right) \right|}_{\text{Term-1}}$$

$$+ \underbrace{\left| \left| (\boldsymbol{e}_a - \boldsymbol{e}_{a'})^{\top} \left( \boldsymbol{\Lambda}_{\boldsymbol{p}_y}^{-1} (\widehat{\boldsymbol{T}}^{\top})^{-1} \boldsymbol{\Lambda}_{\tilde{\boldsymbol{p}}_y} \tilde{\boldsymbol{h}} \right) \right| - \left| (\boldsymbol{e}_a - \boldsymbol{e}_{a'})^{\top} \left( \boldsymbol{\Lambda}_{\boldsymbol{p}_y}^{-1} (\boldsymbol{T}_{k \otimes y}^{\top})^{-1} \boldsymbol{\Lambda}_{\tilde{\boldsymbol{p}}_y} \tilde{\boldsymbol{h}} \right) \right| \right|}_{\text{Term-2}}.$$

Now we upper bound them respectively.

**Term-1:**

$$\left| (e_a - e_{a'})^\top \left( (\Lambda_{\tilde{p}_y}^{-1} - \Lambda_{p_y}^{-1})(\widehat{T}^\top)^{-1} \Lambda_{\tilde{p}_y} \widetilde{h} \right) \right|$$

$$\overset{(a)}{=} \left| (e_a - e_{a'})^\top \left( (\Lambda_{\tilde{p}_y}^{-1} - \Lambda_{p_y}^{-1})(T_{k\otimes y}\widehat{T}^{-1})^\top \Lambda_{p_y} H[:, k \otimes y] \right) \right|$$

$$\overset{(b)}{=} \left| (e_a - e_{a'})^\top \left( (\Lambda_{\tilde{p}_y}^{-1}\Lambda_{p_y} - I)\Lambda_{p_y}^{-1} T_\delta^\top \Lambda_{p_y} H[:, k \otimes y] \right) \right|$$

$$\leq 2 \left\| \Lambda_{\tilde{p}_y}^{-1}\Lambda_{p_y} - I) \right\|_\infty \left\| \Lambda_{p_y}^{-1} \right\|_\infty \| T_\delta \|_1 \| \Lambda_{p_y} H[:, k \otimes y] \|_\infty$$

$$= 2 \left\| \Lambda_{p_y}^{-1} \right\|_\infty \| \Lambda_{p_y} H[:, k \otimes y] \|_\infty \left( \left\| \Lambda_{\tilde{p}_y}^{-1}\Lambda_{p_y} - I) \right\|_\infty \| T_\delta \|_1 \right),$$

where equality $(a)$ holds due to

$$\Lambda_{\tilde{p}_y} \widetilde{H}[:, k \otimes y] = T_{k\otimes y}^\top \Lambda_{p_y} H[:, k \otimes y]$$

and equality $(b)$ holds because we denote the error matrix by $T_\delta$, *i.e.*

$$\widehat{T} = T_\delta^{-1} T_{k\otimes y} \Leftrightarrow T_\delta = T_{k\otimes y}\widehat{T}^{-1}.$$

**Term-2:** Before preceeding, we introduce the Woodbury matrix identity:

$$(A + UCV)^{-1} = A^{-1} - A^{-1}U(C^{-1} + VA^{-1}U)^{-1}VA^{-1}$$

Let $A := T_{k\otimes y}^\top, C = I, V := I, U := \widehat{T}^\top - T_{k\otimes y}^\top$. By Woodbury matrix identity, we have

$$(\widehat{T}^\top)^{-1}$$

$$= (\widehat{T}_{k\otimes y}^\top + (\widehat{T}^\top - T_{k\otimes y}^\top))^{-1}$$

$$= (T_{k\otimes y}^\top)^{-1} - (T_{k\otimes y}^\top)^{-1}(\widehat{T}^\top - T_{k\otimes y}^\top)\left(I + (T_{k\otimes y}^\top)^{-1}(\widehat{T}^\top - T_{k\otimes y}^\top)\right)^{-1}(T_{k\otimes y}^\top)^{-1}$$

Term-2 can be upper bounded as:

$$\left| \left| (e_a - e_{a'})^\top \left( \Lambda_{p_y}^{-1}(\widehat{T}^\top)^{-1}\Lambda_{\tilde{p}_y}\widetilde{h} \right) \right| - \left| (e_a - e_{a'})^\top \left( \Lambda_{p_y}^{-1}(T_{k\otimes y}^\top)^{-1}\Lambda_{\tilde{p}_y}\widetilde{h} \right) \right| \right|$$

$$\overset{(a)}{=} \left| \left| (e_a - e_{a'})^\top \left( \Lambda_{p_y}^{-1}\left( (T_{k\otimes y}^\top)^{-1} - (T_{k\otimes y}^\top)^{-1}(\widehat{T}^\top - T_{k\otimes y}^\top)\left(I + (T_{k\otimes y}^\top)^{-1}(\widehat{T}^\top - T_{k\otimes y}^\top)\right)^{-1}(T_{k\otimes y}^\top)^{-1} \right)\Lambda_{\tilde{p}_y}\widetilde{h} \right) \right| \right.$$

$$\left. - \left| (e_a - e_{a'})^\top \left( \Lambda_{p_y}^{-1}(T_{k\otimes y}^\top)^{-1}\Lambda_{\tilde{p}_y}\widetilde{h} \right) \right| \right|$$

$$\leq \left| (e_a - e_{a'})^\top \left( \Lambda_{p_y}^{-1}(T_{k\otimes y}^\top)^{-1}(\widehat{T}^\top - T_{k\otimes y}^\top)\left(I + (T_{k\otimes y}^\top)^{-1}(\widehat{T}^\top - T_{k\otimes y}^\top)\right)^{-1}(T_{k\otimes y}^\top)^{-1}\Lambda_{\tilde{p}_y}\widetilde{h} \right) \right|$$

$$\overset{(b)}{\leq} \| e_a - e_{a'} \|_1 \left\| \Lambda_{p_y}^{-1}(T_{k\otimes y}^\top)^{-1}(\widehat{T}^\top - T_{k\otimes y}^\top)\left(I + (T_{k\otimes y}^\top)^{-1}(\widehat{T}^\top - T_{k\otimes y}^\top)\right)^{-1}(T_{k\otimes y}^\top)^{-1}\Lambda_{\tilde{p}_y}\widetilde{h} \right\|_\infty$$

$$\leq 2 \| \Lambda_{p_y}^{-1} \|_\infty \left\| (T_{k\otimes y}^\top)^{-1}(\widehat{T}^\top - T_{k\otimes y}^\top)\left(I + (T_{k\otimes y}^\top)^{-1}(\widehat{T}^\top - T_{k\otimes y}^\top)\right)^{-1}(T_{k\otimes y}^\top)^{-1}\Lambda_{\tilde{p}_y}\widetilde{h} \right\|_\infty$$

$$= 2 \| \Lambda_{p_y}^{-1} \|_\infty \left\| \left(I + (T_{k\otimes y}^\top)^{-1}(\widehat{T}^\top - T_{k\otimes y}^\top) - I\right)\left(I + (T_{k\otimes y}^\top)^{-1}(\widehat{T}^\top - T_{k\otimes y}^\top)\right)^{-1}(T_{k\otimes y}^\top)^{-1}\Lambda_{\tilde{p}_y}\widetilde{h} \right\|_\infty$$

$$= 2 \| \Lambda_{p_y}^{-1} \|_\infty \left\| \left[I - \left(I + (T_{k\otimes y}^\top)^{-1}(\widehat{T}^\top - T_{k\otimes y}^\top)\right)^{-1}\right](T_{k\otimes y}^\top)^{-1}\Lambda_{\tilde{p}_y}\widetilde{h} \right\|_\infty$$

$$= 2 \| \Lambda_{p_y}^{-1} \|_\infty \left\| \left(I - T_{k\otimes y}\widehat{T}^{-1}\right)^\top (T_{k\otimes y}^\top)^{-1}\Lambda_{\tilde{p}_y}\widetilde{h} \right\|_\infty$$

$$\overset{(c)}{\leq} 2 \| \Lambda_{p_y}^{-1} \|_\infty \| I - T_\delta \|_1 \left\| (T_{k\otimes y}^\top)^{-1}\Lambda_{\tilde{p}_y}\widetilde{h} \right\|_\infty$$

$$\overset{(d)}{=} 2 \| \Lambda_{p_y}^{-1} \|_\infty \| I - T_\delta \|_1 \| \Lambda_{p_y} H[:, k \otimes y] \|_\infty,$$

where the key steps are:

- (a): Woodbury identity.
- (b): Hölder's inequality.
- (c): $\widehat{T} = T_\delta^{-1} T_{k \otimes y}$ and triangle inequality
- (d):

$$\widetilde{H}[:, k \otimes y] = \Lambda_{\tilde{p}_y}^{-1} T_{k \otimes y}^\top \Lambda_{p_y} H[:, k \otimes y]$$
$$\Leftrightarrow (T_{k \otimes y}^\top)^{-1} \Lambda_{\tilde{p}_y} \widetilde{H}[:, k \otimes y] = \Lambda_{p_y} H[:, k \otimes y].$$

**Wrap-up** Combining the upper bounds of Term-1 and Term-2, we have (recovering full notations)

$$\left| \left| (e_a - e_{a'})^\top \left( \Lambda_{\hat{p}_y}^{-1} (\widehat{T}^\top)^{-1} \Lambda_{\tilde{p}_y} \tilde{h} \right) \right| - \left| (e_a - e_{a'})^\top \left( \Lambda_{p_y}^{-1} (T_{k \otimes y}^\top)^{-1} \Lambda_{\tilde{p}_y} \tilde{h} \right) \right| \right|$$

$$\leq 2 \left\| \Lambda_{p_y}^{-1} \right\|_\infty \left\| \Lambda_{p_y} H[:, k \otimes y] \right\|_\infty \left( \left\| \Lambda_{\hat{p}_y}^{-1} \Lambda_{p_y} - I \right\|_\infty \|T_\delta\|_1 + \|I - T_\delta\|_1 \right)$$

$$= 2 \left\| \Lambda_{p_y}^{-1} \right\|_\infty \left\| \Lambda_{p_y} H[:, k \otimes y] \right\|_\infty \left( \left\| \Lambda_{\hat{p}_y}^{-1} \Lambda_{p_y} - I \right\|_\infty \left\| T_{k \otimes y} \widehat{T}_{k \otimes y}^{-1} \right\|_1 + \left\| I - T_{k \otimes y} \widehat{T}_{k \otimes y}^{-1} \right\|_1 \right).$$

Denote by $\widehat{\Delta}_{k \otimes y}^{\tilde{a}, \tilde{a}'} := |\widehat{H}[\tilde{a}, k \otimes y] - \widehat{H}[\tilde{a}', k \otimes y]|$ the calibrated disparity and $\Delta_{k \otimes y}^{\tilde{a}, \tilde{a}'} := |H[\tilde{a}, k \otimes y] - H[\tilde{a}', k \otimes y]|$ the clean disparity between attributes $\tilde{a}$ and $\tilde{a}'$ in the case when $f(X) = k$ and $Y = y$. We have

$$\left| \widehat{\Delta}^{\mathsf{EOd}}(\widetilde{\mathcal{D}}, f) - \Delta^{\mathsf{EOd}}(\mathcal{D}, f) \right|$$

$$\leq \frac{1}{M(M-1)K^2} \sum_{\tilde{a}, \tilde{a}' \in [M], k, y \in [K]} \left| \widehat{\Delta}_{k \otimes y}^{\tilde{a}, \tilde{a}'} - \Delta_{k \otimes y}^{\tilde{a}, \tilde{a}'} \right|$$

$$\leq \frac{2}{K^2} \sum_{k, y \in [K]} 2 \left\| \Lambda_{p_y}^{-1} \right\|_\infty \left\| \Lambda_{p_y} H[:, k \otimes y] \right\|_\infty \left( \left\| \Lambda_{\hat{p}_y}^{-1} \Lambda_{p_y} - I \right\|_\infty \left\| T_{k \otimes y} \widehat{T}_{k \otimes y}^{-1} \right\|_1 + \left\| I - T_{k \otimes y} \widehat{T}_{k \otimes y}^{-1} \right\|_1 \right).$$

The above inequality can be generalized to DP by dropping dependency on $y$ and to EOp by requiring $k = 1$ and $y = 1$.

$\square$

### B.5 PROOF FOR COROLLARY 2

*Proof.* Consider DP. Denote by $H[:, k = 1] = [h, h']^\top$. We know $\delta = |h - h'|/2 = \Delta^{\mathsf{DP}}(\mathcal{D}, f)/2$. Suppose $p \leq 1/2$, $\left\| \Lambda_p^{-1} \right\|_\infty = 1/p$ and

$$\|\Lambda_p H[:, k]\|_\infty = \max(ph, (1-p)h').$$

Recall

$$\varepsilon(\widehat{T}_k, \hat{p}) := \|\Lambda_{\hat{p}}^{-1} \Lambda_p - I\|_1 \|T_k \widehat{T}_k^{-1}\|_1 + \|I - T_k \widehat{T}_k^{-1}\|_1.$$

By requiring the error upper bound in Theorem 3 less than the exact error in Corollary 1, we have (when $k = 1$)

$$2 \left\| \Lambda_p^{-1} \right\|_\infty \|\Lambda_p H[:, k]\|_\infty \varepsilon(\widehat{T}_k, \hat{p}) 2 \leq \delta \cdot (e_1 + e_2)$$
$$\Leftrightarrow \varepsilon(\widehat{T}_k, \hat{p}) \leq \frac{\delta \cdot (e_1 + e_2)}{\left\| \Lambda_p^{-1} \right\|_\infty \|\Lambda_p H[:, k]\|_\infty}$$
$$\Leftrightarrow \varepsilon(\widehat{T}_k, \hat{p}) \leq \frac{\delta \cdot (e_1 + e_2)}{\max(h, (1-p)h'/p)}.$$

If $p = 1/2$, noting $\max(h, h') = (|h + h'| + |h - h'|)/2$, we further have (when $k = 1$)

$$\varepsilon(\widehat{T}_k, \hat{p}) \leq \frac{|h - h'| \cdot (e_1 + e_2)}{|h - h'| + |h + h'|} = \frac{e_1 + e_2}{1 + \frac{h + h'}{|h - h'|}} = \frac{e_1 + e_2}{1 + \frac{h + h'}{\Delta^{\mathsf{DP}}(\mathcal{D}, f)}}.$$

To make the above equality holds for all $k \in \{1, 2\}$, we have

$$\varepsilon(\widehat{\boldsymbol{T}}_k, \hat{\boldsymbol{p}}) \leq \max_{k' \in \{1,2\}} \frac{e_1 + e_2}{1 + \frac{\|\boldsymbol{H}[:,k']\|_1}{\Delta^{\mathsf{DP}}(\mathcal{D}, f)}}, \forall k \in \{1, 2\}.$$

$\square$

## C  MORE DISCUSSIONS ON TRANSITION MATRIX ESTIMATORS

### C.1  HOC

HOC (Zhu et al., 2021b) relies on checking the agreements and disagreements among three noisy attributes of one feature. For example, given a three-tuple $(\tilde{a}_n^1, \tilde{a}_n^2, \tilde{a}_n^3)$, each noisy attribute may agree or disagree with the others. This consensus pattern encodes the information of noise transition matrix $\boldsymbol{T}$. Suppose $(\tilde{a}_n^1, \tilde{a}_n^2, \tilde{a}_n^3)$ are drawn from random variables $(\tilde{A}^1, \tilde{A}^2, \tilde{A}^3)$ satisfying Assumption 2. Denote by

$$e_1 = \mathbb{P}(\widetilde{A}^1 = 2 | A^1 = 1) = \mathbb{P}(\widetilde{A}^2 = 2 | A^2 = 1) = \mathbb{P}(\widetilde{A}^3 = 2 | A^3 = 1)$$

and

$$e_2 = \mathbb{P}(\widetilde{A}^1 = 1 | A^1 = 2) = \mathbb{P}(\widetilde{A}^2 = 1 | A^2 = 2) = \mathbb{P}(\widetilde{A}^3 = 1 | A^3 = 2).$$

Note $A_1 = A_2 = A_3$. We have:

- First order equations:

$$\mathbb{P}(\tilde{A}^1 = 1) = \mathbb{P}(A^1 = 1) \cdot (1 - e_1) + \mathbb{P}(A^1 = 2) \cdot e_2$$
$$\mathbb{P}(\tilde{A}^1 = 2) = \mathbb{P}(A^1 = 1) \cdot e_1 + \mathbb{P}(A^1 = 2) \cdot (1 - e_2)$$

- Second order equations:

$$\mathbb{P}(\tilde{A}^1 = 1, \tilde{A}^2 = 1) = \mathbb{P}(\tilde{A}^1 = 1, \tilde{A}^2 = 1 | A_1 = 1) \cdot \mathbb{P}(A^1 = 1) + \mathbb{P}(\tilde{A}^1 = 1, \tilde{A}^2 = 1 | A_1 = 2) \cdot \mathbb{P}(A^1 = 2)$$
$$= (1 - e_1)^2 \cdot \mathbb{P}(A^1 = 1) + e_2^2 \cdot \mathbb{P}(A^1 = 2).$$

Similarly,

$$\mathbb{P}(\tilde{A}^1 = 1, \tilde{A}^2 = 2) = (1 - e_1)e_1 \cdot \mathbb{P}(A^1 = 1) + e_2(1 - e_2) \cdot \mathbb{P}(A^1 = 2)$$
$$\mathbb{P}(\tilde{A}^1 = 2, \tilde{A}^2 = 1) = (1 - e_1)e_1 \cdot \mathbb{P}(A^1 = 1) + e_2(1 - e_2) \cdot \mathbb{P}(A^1 = 2)$$
$$\mathbb{P}(\tilde{A}^1 = 2, \tilde{A}^2 = 2) = e_1^2 \cdot \mathbb{P}(A^1 = 1) + (1 - e_2)^2 \cdot \mathbb{P}(A^1 = 2).$$

- Third order equations:

$$\mathbb{P}(\tilde{A}^1 = 1, \tilde{A}^2 = 1, \tilde{A}^3 = 1) = (1 - e_1)^3 \cdot \mathbb{P}(A^1 = 1) + e_2^3 \cdot \mathbb{P}(A^1 = 2)$$
$$\mathbb{P}(\tilde{A}^1 = 1, \tilde{A}^2 = 1, \tilde{A}^3 = 2) = (1 - e_1)^2 e_1 \cdot \mathbb{P}(A^1 = 1) + (1 - e_2)e_2^2 \cdot \mathbb{P}(A^1 = 2)$$
$$\mathbb{P}(\tilde{A}^1 = 1, \tilde{A}^2 = 2, \tilde{A}^3 = 2) = (1 - e_1)e_1^2 \cdot \mathbb{P}(A^1 = 1) + (1 - e_2)^2 e_2 \cdot \mathbb{P}(A^1 = 2)$$
$$\mathbb{P}(\tilde{A}^1 = 1, \tilde{A}^2 = 2, \tilde{A}^3 = 1) = (1 - e_1)^2 e_1 \cdot \mathbb{P}(A^1 = 1) + (1 - e_2)e_2^2 \cdot \mathbb{P}(A^1 = 2)$$
$$\mathbb{P}(\tilde{A}^1 = 2, \tilde{A}^2 = 1, \tilde{A}^3 = 1) = (1 - e_1)^2 e_1 \cdot \mathbb{P}(A^1 = 1) + (1 - e_2)e_2^2 \cdot \mathbb{P}(A^1 = 2)$$
$$\mathbb{P}(\tilde{A}^1 = 2, \tilde{A}^2 = 1, \tilde{A}^3 = 2) = (1 - e_1)e_1^2 \cdot \mathbb{P}(A^1 = 1) + (1 - e_2)^2 e_2 \cdot \mathbb{P}(A^1 = 2)$$
$$\mathbb{P}(\tilde{A}^1 = 2, \tilde{A}^2 = 2, \tilde{A}^3 = 1) = (1 - e_1)e_1^2 \cdot \mathbb{P}(A^1 = 1) + (1 - e_2)^2 e_2 \cdot \mathbb{P}(A^1 = 2)$$
$$\mathbb{P}(\tilde{A}^1 = 2, \tilde{A}^2 = 2, \tilde{A}^3 = 2) = e_1^3 \cdot \mathbb{P}(A^1 = 1) + (1 - e_2)^3 \cdot \mathbb{P}(A^1 = 2).$$

With the above equations, we can count the frequency of each pattern (LHS) as $(\hat{c}^{[1]}, \hat{c}^{[2]}, \hat{c}^{[3]})$ and solve the equations. See the key steps summarized in Algorithm 3.

---

**Algorithm 3** Key Steps of HOC

---

1: **Input:** A set of three-tuples: $\{(\tilde{a}_n^1, \tilde{a}_n^2, \tilde{a}_n^3)|n \in [N]\}$
2: $(\hat{c}^{[1]}, \hat{c}^{[2]}, \hat{c}^{[3]}) \leftarrow \texttt{CountFreq}(\{(\tilde{a}_n^1, \tilde{a}_n^2, \tilde{a}_n^3)|n \in [N]\})$ *// Count 1st, 2nd, and 3rd-order patterns*
3: Find $T$ such that match the counts $(\hat{c}^{[1]}, \hat{c}^{[2]}, \hat{c}^{[3]})$ *// Solve equations*

---

Table 5: Normalized error ($\times 100$) of a learning-centric estimator.

| Method | DP Global | DP Local | EOd Global | EOd Local | EOp Global | EOp Local |
|---|---|---|---|---|---|---|
| Base | 15.33 | / | 4.11 | / | 2.82 | / |
| Northcutt et al. (2021) | 15.37 | 15.49 | 4.07 | 4.02 | 2.86 | 2.95 |

Table 6: Disparities in the COMPAS dataset

| COMPAS | True | | | Uncalibrated Noisy | | |
|---|---|---|---|---|---|---|
| | DP | EOd | EOp | DP | EOd | EOp |
| tree | 0.2424 | 0.2013 | 0.2541 | 0.1362 | 0.1090 | 0.1160 |
| forest | 0.2389 | 0.1947 | 0.2425 | 0.1346 | 0.1059 | 0.1120 |
| boosting | 0.2424 | 0.2013 | 0.2541 | 0.1362 | 0.1090 | 0.1160 |
| SVM | 0.2535 | 0.2135 | 0.2577 | 0.1252 | 0.0988 | 0.1038 |
| logit | 0.2000 | 0.1675 | 0.2278 | 0.1169 | 0.0950 | 0.1120 |
| nn | 0.2318 | 0.1913 | 0.2359 | 0.1352 | 0.1084 | 0.1073 |
| compas_score | 0.2572 | 0.2217 | 0.2586 | 0.1511 | 0.1276 | 0.1324 |

## C.2 OTHER ESTIMATORS THAT REQUIRE TRAINING

The other estimators in the noisy label literature mainly focus on training a new model to fit the noisy data distribution. The intuition is that the new model has the ability to distinguish between true attributes and wrong attributes. In other words, they believe the prediction of new model is close to the true attributes. It is useful when the noise in attributes are random. However, this intuitions is hardly true in our setting since we need to train a new model to learn the noisy attributes given by an auxiliary model, which are deterministic. One caveat of this approach is that the new model is likely to fit the auxiliary model when both the capacity of the new model and the amount of data are sufficient, leading to a trivial transition matrix estimate that is an identity matrix, i.e., $T = I$. In this case, the performance is close to Base. We reproduce Northcutt et al. (2021) follow the setting in Table 2 and summarize the result in Table 5, which verifies that the performance of this kind of approach is close to Base.

## D MORE EXPERIMENTAL RESULTS

### D.1 MORE TABLES FOR THE COMPAS DATASET

We have two tables in this subsection.

- Table 6 shows the raw disparities measured on the COMPAS dataset.
- Table 7 is the full version of Table 1.

### D.2 MORE TABLES FOR THE CELEBA DATASET

We have three tables in this subsection.

- Table 8 is the full version of Table 2.
- Table 9 is the full version of Table 3.
- Table 10 is similar to Table 9, but the error metric is changed to Improvement defined in Section 5.1.

### D.3 MORE DISCUSSIONS

We cross-reference different tables and show several takeaway messages. Note some of them have been introduced in the main paper.

**Uncalibrated measurement is sensitive to noise rates and raw disparity**     The target models $f$ trained on the COMPAS dataset are usually biased (details in Table 6), and the auxiliary models $g$ are inaccurate (accuracy 68.85% in binary classifications), while $f$ is almost not biased in EOd and EOp (details in Table 8) and $g$ is accurate (accuracy 92.55%). As a result, all the three types of directly measured fairness metrics (Base) have large normalized errors ($\sim$ 40–60%) as in Table 1, a moderate normalized error in DP (15.33%), and small normalized errors in EOd (4.11%) and EOp (2.82%) as in Table 2, which is consistent with our results in Theorem 1 and Corollary 1.

**Local vs. Global**     Table 1 also shows our Global method works consistently better than the Local method, while Table 2 has the reversed result. Intuitively, when the auxiliary models are highly inaccurate (accuracy 68.85%), Assumptions 1–2 for implementing HOC may not hold well in every local dataset, inducing large estimation errors in local estimates and unstable calibrations. On the contrary, when the auxiliary models are accurate (92.55% accuracy in Table 2), Assumption 1 always hold and most instances will satisfy Assumption 2 if we carefully choose the other two auxiliary models $g_2$ and $g_3$ given $g_1$, then Local will outperform since it can achieve 0 error if both assumptions perfectly hold and Global induces extra error due to approximation. Note Table 3 shows Local is still statistically better than Global when the noise rate is high. This is because the extra random flipping follows Assumption 2 and the estimation error of Local is not improved significantly. Therefore, we prefer Local when the original auxiliary model is accurate and Global to stabilize the calibration otherwise.

**When our method is better**     Both Table 1, DP in Table 2, and Table 3 show our method is significantly better than both baselines, where the noise rates of $g$ are moderate to high (e.g., $\geq 15\%$) or $f$ is biased (e.g., $\geq 0.1$). This observation is also consistent with our result in Corollary 2. In other cases when both the noise rate and original disparity are low, our calibration may not be perfect compared with others without calibration, e.g., EOp in Table 2. However, the raw error of EOp is sufficiently small ($< 0.01$) for all approaches, indicating the absolute performance of our method is not bad although it fails to be better than others.

Table 7: Performance on the COMPAS dataset. The method with minimal normalized error is **bold**.

| COMPAS | DP Normalized Error (%) ↓ | | | | EOd Normalized Error (%) ↓ | | | | EOp Normalized Error (%) ↓ | | | |
|---|---|---|---|---|---|---|---|---|---|---|---|---|
| | Base | Soft | Global | Local | Base | Soft | Global | Local | Base | Soft | Global | Local |
| tree | 43.82 | 61.26 | **22.29** | 39.81 | 45.86 | 63.96 | **23.09** | 42.81 | 54.36 | 70.15 | **13.27** | 49.49 |
| forest | 43.68 | 60.30 | **19.65** | 44.14 | 45.60 | 62.85 | **18.56** | 44.04 | 53.83 | 69.39 | **17.51** | 63.62 |
| boosting | 43.82 | 61.26 | **22.29** | 44.64 | 45.86 | 63.96 | **23.25** | 49.08 | 54.36 | 70.15 | **13.11** | 54.67 |
| SVM | 50.61 | 66.50 | **30.95** | 42.00 | 53.72 | 69.69 | **32.46** | 47.39 | 59.70 | 71.12 | **29.29** | 51.31 |
| logit | 41.54 | 60.78 | **16.98** | 35.69 | 43.26 | 63.15 | **21.42** | 31.91 | 50.86 | 65.04 | **14.90** | 26.27 |
| nn | 41.69 | 60.55 | **19.48** | 34.22 | 43.34 | 62.99 | **19.30** | 43.24 | 54.50 | 68.50 | **14.20** | 59.95 |
| compas_score | 41.28 | 58.34 | **11.24** | 14.66 | 42.43 | 59.79 | **11.80** | 18.65 | 48.78 | 62.24 | **5.78** | 23.80 |

| | DP Raw Disparity ↓ | | | | EOd Raw Disparity ↓ | | | | EOp Raw Disparity ↓ | | | |
|---|---|---|---|---|---|---|---|---|---|---|---|---|
| tree | 0.1362 | 0.0939 | 0.1884 | 0.1459 | 0.1090 | 0.0726 | 0.1548 | 0.1151 | 0.1160 | 0.0759 | 0.2204 | 0.1283 |
| forest | 0.1345 | 0.0948 | 0.1919 | 0.1334 | 0.1059 | 0.0723 | 0.1586 | 0.1090 | 0.1120 | 0.0743 | 0.2001 | 0.0882 |
| boosting | 0.1362 | 0.0939 | 0.1884 | 0.1342 | 0.1090 | 0.0726 | 0.1545 | 0.1025 | 0.1160 | 0.0759 | 0.2208 | 0.1152 |
| SVM | 0.1252 | 0.0849 | 0.1750 | 0.1470 | 0.0988 | 0.0647 | 0.1442 | 0.1123 | 0.1038 | 0.0744 | 0.1822 | 0.1255 |
| logit | 0.1169 | 0.0784 | 0.1660 | 0.1286 | 0.0950 | 0.0617 | 0.1316 | 0.1140 | 0.1120 | 0.0797 | 0.1939 | 0.1680 |
| nn | 0.1352 | 0.0915 | 0.1867 | 0.1525 | 0.1084 | 0.0708 | 0.1544 | 0.1086 | 0.1073 | 0.0743 | 0.2024 | 0.0945 |
| compas_score | 0.1510 | 0.1072 | 0.2283 | 0.2195 | 0.1276 | 0.0891 | 0.1955 | 0.1803 | 0.1324 | 0.0976 | 0.2436 | 0.1970 |

| | DP Raw Error ↓ | | | | EOd Raw Error ↓ | | | | EOp Raw Error ↓ | | | |
|---|---|---|---|---|---|---|---|---|---|---|---|---|
| tree | 0.1062 | 0.1485 | 0.0540 | 0.0965 | 0.0923 | 0.1288 | 0.0465 | 0.0862 | 0.1381 | 0.1782 | 0.0337 | 0.1257 |
| forest | 0.1043 | 0.1440 | 0.0469 | 0.1054 | 0.0888 | 0.1224 | 0.0361 | 0.0858 | 0.1306 | 0.1683 | 0.0425 | 0.1543 |
| boosting | 0.1062 | 0.1485 | 0.0540 | 0.1082 | 0.0923 | 0.1288 | 0.0468 | 0.0988 | 0.1381 | 0.1782 | 0.0333 | 0.1389 |
| SVM | 0.1283 | 0.1685 | 0.0785 | 0.1064 | 0.1147 | 0.1488 | 0.0693 | 0.1012 | 0.1538 | 0.1833 | 0.0755 | 0.1322 |
| logit | 0.0831 | 0.1215 | 0.0340 | 0.0714 | 0.0724 | 0.1057 | 0.0359 | 0.0534 | 0.1159 | 0.1482 | 0.0339 | 0.0598 |
| nn | 0.0966 | 0.1404 | 0.0452 | 0.0793 | 0.0829 | 0.1205 | 0.0369 | 0.0827 | 0.1286 | 0.1616 | 0.0335 | 0.1414 |
| compas_score | 0.1062 | 0.1500 | 0.0289 | 0.0377 | 0.0941 | 0.1325 | 0.0261 | 0.0413 | 0.1261 | 0.1609 | 0.0150 | 0.0615 |

| | DP Improvement (%) ↑ | | | | EOd Improvement (%) ↑ | | | | EOp Improvement (%) ↑ | | | |
|---|---|---|---|---|---|---|---|---|---|---|---|---|
| tree | 0.00 | -39.79 | 49.15 | 9.15 | 0.00 | -39.48 | 49.65 | 6.64 | 0.00 | -29.05 | 75.60 | 8.96 |
| forest | 0.00 | -38.05 | 55.01 | -1.06 | 0.00 | -37.83 | 59.30 | 3.42 | 0.00 | -28.89 | 67.47 | -18.18 |
| boosting | 0.00 | -39.79 | 49.15 | -1.87 | 0.00 | -39.48 | 49.30 | -7.04 | 0.00 | -29.05 | 75.89 | -0.57 |
| SVM | 0.00 | -31.40 | 38.83 | 17.02 | 0.00 | -29.72 | 39.57 | 11.78 | 0.00 | -19.12 | 50.93 | 14.05 |
| logit | 0.00 | -46.30 | 59.12 | 14.08 | 0.00 | -45.98 | 50.47 | 26.24 | 0.00 | -27.87 | 70.70 | 48.35 |
| nn | 0.00 | -45.23 | 53.27 | 17.93 | 0.00 | -45.34 | 55.47 | 0.23 | 0.00 | -25.69 | 73.94 | -10.01 |
| compas_score | 0.00 | -41.33 | 72.77 | 64.48 | 0.00 | -40.92 | 72.20 | 56.04 | 0.00 | -27.59 | 88.15 | 51.21 |

Table 8: Performance on the CelebA dataset. The method with minimal normalized error is **bold**.

| CelebA | DP Normalized Error (%) ↓ | | | | EOd Normalized Error (%) ↓ | | | | EOp Normalized Error (%) ↓ | | | |
|---|---|---|---|---|---|---|---|---|---|---|---|---|
| | Base | Soft | Global | Local | Base | Soft | Global | Local | Base | Soft | Global | Local |
| Facenet | 15.33 | 12.54 | 22.17 | **10.89** | 4.11 | 6.46 | 7.54 | **0.26** | 2.82 | **0.34** | 12.22 | 2.93 |
| Facenet512 | 15.33 | 12.54 | 21.70 | **7.26** | 4.11 | 6.46 | 4.85 | **0.52** | 2.82 | **0.34** | 11.80 | 3.24 |
| OpenFace | 15.33 | 12.54 | 10.31 | **9.39** | **4.11** | 6.46 | 10.43 | 5.03 | 2.82 | **0.34** | 0.56 | 0.93 |
| ArcFace | 15.33 | 12.54 | 19.59 | **9.69** | 4.11 | 6.46 | 5.72 | **0.23** | 2.82 | **0.34** | 11.16 | 3.85 |
| Dlib | 15.33 | 12.54 | 15.09 | **5.30** | **4.11** | 6.46 | 4.87 | 4.25 | 2.82 | **0.34** | 9.74 | 2.32 |
| SFace | 15.33 | 12.54 | 17.00 | **4.77** | 4.11 | 6.46 | 4.04 | **3.91** | 2.82 | **0.34** | 9.36 | 3.28 |

| | DP Raw Disparity ↓ | | | | EOd Raw Disparity ↓ | | | | EOp Raw Disparity ↓ | | | |
|---|---|---|---|---|---|---|---|---|---|---|---|---|
| Facenet | 0.1522 | 0.1485 | 0.1612 | 0.1464 | 0.0316 | 0.0309 | 0.0355 | 0.0331 | 0.0573 | 0.0559 | 0.0625 | 0.0573 |
| Facenet512 | 0.1522 | 0.1485 | 0.1606 | 0.1416 | 0.0316 | 0.0309 | 0.0346 | 0.0328 | 0.0573 | 0.0559 | 0.0623 | 0.0575 |
| OpenFace | 0.1522 | 0.1485 | 0.1456 | 0.1444 | 0.0316 | 0.0309 | 0.0295 | 0.0313 | 0.0573 | 0.0559 | 0.0554 | 0.0552 |
| ArcFace | 0.1522 | 0.1485 | 0.1578 | 0.1448 | 0.0316 | 0.0309 | 0.0349 | 0.0329 | 0.0573 | 0.0559 | 0.0619 | 0.0578 |
| Dlib | 0.1522 | 0.1485 | 0.1519 | 0.1390 | 0.0316 | 0.0309 | 0.0346 | 0.0316 | 0.0573 | 0.0559 | 0.0611 | 0.0544 |
| SFace | 0.1522 | 0.1485 | 0.1544 | 0.1383 | 0.0316 | 0.0309 | 0.0343 | 0.0317 | 0.0573 | 0.0559 | 0.0609 | 0.0539 |

| | DP Raw Error ↓ | | | | EOd Raw Error ↓ | | | | EOp Raw Error ↓ | | | |
|---|---|---|---|---|---|---|---|---|---|---|---|---|
| Facenet | 0.0202 | 0.0165 | 0.0293 | 0.0144 | 0.0014 | 0.0021 | 0.0025 | 0.0001 | 0.0016 | 0.0002 | 0.0068 | 0.0016 |
| Facenet512 | 0.0202 | 0.0165 | 0.0286 | 0.0096 | 0.0014 | 0.0021 | 0.0016 | 0.0002 | 0.0016 | 0.0002 | 0.0066 | 0.0018 |
| OpenFace | 0.0202 | 0.0165 | 0.0136 | 0.0124 | 0.0014 | 0.0021 | 0.0034 | 0.0017 | 0.0016 | 0.0002 | 0.0003 | 0.0005 |
| ArcFace | 0.0202 | 0.0165 | 0.0259 | 0.0128 | 0.0014 | 0.0021 | 0.0019 | 0.0001 | 0.0016 | 0.0002 | 0.0062 | 0.0021 |
| Dlib | 0.0202 | 0.0165 | 0.0199 | 0.0070 | 0.0014 | 0.0021 | 0.0016 | 0.0014 | 0.0016 | 0.0002 | 0.0054 | 0.0013 |
| SFace | 0.0202 | 0.0165 | 0.0224 | 0.0063 | 0.0014 | 0.0021 | 0.0013 | 0.0013 | 0.0016 | 0.0002 | 0.0052 | 0.0018 |

| | DP Improvement (%) ↑ | | | | EOd Improvement (%) ↑ | | | | EOp Improvement (%) ↑ | | | |
|---|---|---|---|---|---|---|---|---|---|---|---|---|
| Facenet | 0.00 | 18.22 | -44.58 | 28.99 | 0.00 | -57.38 | -83.62 | 93.64 | 0.00 | 88.05 | -333.85 | -3.97 |
| Facenet512 | 0.00 | 18.22 | -41.50 | 52.65 | 0.00 | -57.38 | -18.15 | 87.29 | 0.00 | 88.05 | -319.18 | -15.09 |
| OpenFace | 0.00 | 18.22 | 32.76 | 38.75 | 0.00 | -57.38 | -154.12 | -22.45 | 0.00 | 88.05 | 80.03 | 67.15 |
| ArcFace | 0.00 | 18.22 | -27.78 | 36.78 | 0.00 | -57.38 | -39.45 | 94.31 | 0.00 | 88.05 | -296.25 | -36.65 |
| Dlib | 0.00 | 18.22 | 1.56 | 65.46 | 0.00 | -57.38 | -18.55 | -3.43 | 0.00 | 88.05 | -245.95 | 17.61 |
| SFace | 0.00 | 18.22 | -10.87 | 68.87 | 0.00 | -57.38 | 1.61 | 4.85 | 0.00 | 88.05 | -232.48 | -16.46 |

Table 9: Normalized Error on CelebA with different noise rates

| CelebA | DP Normalized Error (%) ↓ | | | | EOd Normalized Error (%) ↓ | | | | EOp Normalized Error (%) ↓ | | | |
|---|---|---|---|---|---|---|---|---|---|---|---|---|
| | Base | Soft | Global | Local | Base | Soft | Global | Local | Base | Soft | Global | Local |
| Facenet [0.0, 0.0] | 15.33 | 12.54 | 22.17 | 10.89 | 4.11 | 6.46 | 7.54 | 0.26 | 2.82 | 0.34 | 12.22 | 2.93 |
| Facenet [0.2, 0.0] | 7.39 | 11.65 | 20.75 | 10.82 | 25.05 | 26.99 | 9.87 | 6.63 | 24.69 | 27.27 | 11.55 | 2.77 |
| Facenet [0.2, 0.2] | 30.24 | 31.57 | 24.27 | 8.45 | 44.71 | 46.36 | 15.10 | 3.99 | 37.67 | 38.77 | 21.79 | 16.73 |
| Facenet [0.4, 0.2] | 51.37 | 54.56 | 20.12 | 20.66 | 62.94 | 65.10 | 3.45 | 3.67 | 56.53 | 58.73 | 15.75 | 2.70 |
| Facenet [0.4, 0.4] | 77.82 | 78.39 | 8.76 | 21.94 | 79.36 | 80.10 | 51.32 | 148.05 | 78.39 | 79.62 | 71.38 | 146.20 |
| Facenet512 [0.0, 0.0] | 15.33 | 12.54 | 21.70 | 7.26 | 4.11 | 6.46 | 4.85 | 0.52 | 2.82 | 0.34 | 11.80 | 3.24 |
| Facenet512 [0.2, 0.0] | 7.37 | 11.65 | 20.58 | 5.05 | 25.06 | 26.99 | 6.43 | 0.10 | 24.69 | 27.27 | 11.11 | 1.07 |
| Facenet512 [0.2, 0.2] | 30.21 | 31.57 | 24.25 | 13.10 | 44.73 | 46.36 | 11.26 | 9.04 | 37.67 | 38.77 | 20.94 | 27.98 |
| Facenet512 [0.4, 0.2] | 51.32 | 54.56 | 19.42 | 10.47 | 62.90 | 65.10 | 11.09 | 19.15 | 56.51 | 58.73 | 23.86 | 23.55 |
| Facenet512 [0.4, 0.4] | 77.76 | 78.39 | 9.41 | 19.80 | 79.31 | 80.10 | 24.49 | 8.02 | 78.35 | 79.62 | 10.61 | 5.71 |
| OpenFace [0.0, 0.0] | 15.33 | 12.54 | 10.31 | 9.39 | 4.11 | 6.46 | 10.43 | 5.03 | 2.82 | 0.34 | 0.56 | 0.93 |
| OpenFace [0.2, 0.0] | 7.39 | 11.65 | 8.93 | 6.60 | 25.05 | 26.99 | 9.86 | 13.01 | 24.69 | 27.27 | 1.08 | 10.96 |
| OpenFace [0.2, 0.2] | 30.24 | 31.57 | 13.32 | 21.46 | 44.74 | 46.36 | 7.56 | 15.88 | 37.69 | 38.77 | 5.90 | 7.40 |
| OpenFace [0.4, 0.2] | 51.39 | 54.56 | 10.66 | 25.16 | 62.96 | 65.10 | 6.47 | 24.94 | 56.55 | 58.73 | 6.11 | 47.12 |
| OpenFace [0.4, 0.4] | 77.84 | 78.39 | 1.60 | 117.27 | 79.38 | 80.10 | 34.00 | 19.47 | 78.41 | 79.62 | 37.42 | 31.99 |
| ArcFace [0.0, 0.0] | 15.33 | 12.54 | 19.59 | 9.69 | 4.11 | 6.46 | 5.72 | 0.23 | 2.82 | 0.34 | 11.16 | 3.85 |
| ArcFace [0.2, 0.0] | 7.39 | 11.65 | 17.74 | 7.74 | 25.05 | 26.99 | 6.18 | 1.82 | 24.69 | 27.27 | 8.81 | 3.37 |
| ArcFace [0.2, 0.2] | 30.19 | 31.57 | 21.77 | 8.97 | 44.77 | 46.36 | 12.12 | 18.91 | 37.69 | 38.77 | 21.19 | 17.99 |
| ArcFace [0.4, 0.2] | 51.32 | 54.56 | 17.33 | 44.52 | 62.91 | 65.10 | 14.66 | 29.74 | 56.53 | 58.73 | 24.39 | 4.92 |
| ArcFace [0.4, 0.4] | 77.79 | 78.39 | 8.38 | 84.37 | 79.34 | 80.10 | 8.31 | 165.03 | 78.39 | 79.62 | 16.98 | 62.34 |
| Dlib [0.0, 0.0] | 15.33 | 12.54 | 15.09 | 5.30 | 4.11 | 6.46 | 4.87 | 4.25 | 2.82 | 0.34 | 9.74 | 2.32 |
| Dlib [0.2, 0.0] | 7.35 | 11.65 | 14.39 | 1.06 | 25.07 | 26.99 | 3.78 | 2.63 | 24.69 | 27.27 | 7.09 | 2.36 |
| Dlib [0.2, 0.2] | 30.23 | 31.57 | 16.78 | 1.95 | 44.77 | 46.36 | 9.50 | 11.28 | 37.72 | 38.77 | 15.88 | 22.43 |
| Dlib [0.4, 0.2] | 51.40 | 54.56 | 12.83 | 17.69 | 62.96 | 65.10 | 10.34 | 11.47 | 56.57 | 58.73 | 18.90 | 11.17 |
| Dlib [0.4, 0.4] | 77.84 | 78.39 | 0.46 | 96.58 | 79.38 | 80.10 | 7.99 | 86.36 | 78.41 | 79.62 | 8.45 | 14.78 |
| SFace [0.0, 0.0] | 15.33 | 12.54 | 17.00 | 4.77 | 4.11 | 6.46 | 4.04 | 3.91 | 2.82 | 0.34 | 9.36 | 3.28 |
| SFace [0.2, 0.0] | 7.41 | 11.65 | 15.18 | 1.94 | 25.04 | 26.99 | 3.31 | 8.82 | 24.69 | 27.27 | 7.24 | 13.05 |
| SFace [0.2, 0.2] | 30.22 | 31.57 | 18.16 | 20.95 | 44.72 | 46.36 | 4.58 | 20.93 | 37.67 | 38.77 | 11.55 | 34.72 |
| SFace [0.4, 0.2] | 51.35 | 54.56 | 14.72 | 48.96 | 62.92 | 65.10 | 2.95 | 68.93 | 56.51 | 58.73 | 15.22 | 68.85 |
| SFace [0.4, 0.4] | 77.78 | 78.39 | 3.37 | 31.25 | 79.33 | 80.10 | 21.56 | 178.21 | 78.37 | 79.62 | 20.03 | 86.59 |

Table 10: Improvement on CelebA with different noise rates

| CelebA | DP Improvement (%) ↑ | | | | EOd Improvement (%) ↑ | | | | EOp Improvement (%) ↑ | | | |
|---|---|---|---|---|---|---|---|---|---|---|---|---|
| | Base | Soft | Global | Local | Base | Soft | Global | Local | Base | Soft | Global | Local |
| Facenet [0.0, 0.0] | 0.00 | 18.22 | -44.58 | 28.99 | 0.00 | -57.38 | -83.62 | 93.64 | 0.00 | 88.05 | -333.85 | -3.97 |
| Facenet [0.2, 0.0] | 0.00 | -57.70 | -180.88 | -46.45 | 0.00 | -7.75 | 60.60 | 73.52 | 0.00 | -10.44 | 53.24 | 88.80 |
| Facenet [0.2, 0.2] | 0.00 | -4.39 | 19.75 | 72.05 | 0.00 | -3.69 | 66.22 | 91.07 | 0.00 | -2.92 | 42.17 | 55.58 |
| Facenet [0.4, 0.2] | 0.00 | -6.20 | 60.83 | 59.79 | 0.00 | -3.44 | 94.51 | 94.17 | 0.00 | -3.90 | 72.13 | 95.23 |
| Facenet [0.4, 0.4] | 0.00 | -0.73 | 88.74 | 71.81 | 0.00 | -0.94 | 35.33 | -86.56 | 0.00 | -1.57 | 8.94 | -86.50 |
| Facenet512 [0.0, 0.0] | 0.00 | 18.22 | -41.50 | 52.65 | 0.00 | -57.38 | -18.15 | 87.29 | 0.00 | 88.05 | -319.18 | -15.09 |
| Facenet512 [0.2, 0.0] | 0.00 | -58.10 | -179.28 | 31.43 | 0.00 | -7.70 | 74.32 | 99.58 | 0.00 | -10.44 | 54.98 | 95.68 |
| Facenet512 [0.2, 0.2] | 0.00 | -4.51 | 19.72 | 56.64 | 0.00 | -3.64 | 74.81 | 79.78 | 0.00 | -2.92 | 44.40 | 25.73 |
| Facenet512 [0.4, 0.2] | 0.00 | -6.32 | 62.17 | 79.60 | 0.00 | -3.50 | 82.37 | 69.55 | 0.00 | -3.94 | 57.78 | 58.33 |
| Facenet512 [0.4, 0.4] | 0.00 | -0.81 | 87.90 | 74.54 | 0.00 | -1.00 | 69.12 | 89.89 | 0.00 | -1.63 | 86.45 | 92.71 |
| OpenFace [0.0, 0.0] | 0.00 | 18.22 | 32.76 | 38.75 | 0.00 | -57.38 | -154.12 | -22.45 | 0.00 | 88.05 | 80.03 | 67.15 |
| OpenFace [0.2, 0.0] | 0.00 | -57.70 | -20.83 | 10.69 | 0.00 | -7.75 | 60.65 | 48.05 | 0.00 | -10.44 | 95.65 | 55.62 |
| OpenFace [0.2, 0.2] | 0.00 | -4.38 | 55.97 | 29.06 | 0.00 | -3.62 | 83.11 | 64.51 | 0.00 | -2.86 | 84.35 | 80.38 |
| OpenFace [0.4, 0.2] | 0.00 | -6.16 | 79.25 | 51.05 | 0.00 | -3.41 | 89.72 | 60.39 | 0.00 | -3.86 | 89.19 | 16.67 |
| OpenFace [0.4, 0.4] | 0.00 | -0.71 | 97.94 | -50.65 | 0.00 | -0.92 | 57.17 | 75.47 | 0.00 | -1.54 | 52.28 | 59.20 |
| ArcFace [0.0, 0.0] | 0.00 | 18.22 | -27.78 | 36.78 | 0.00 | -57.38 | -39.45 | 94.31 | 0.00 | 88.05 | -296.25 | -36.65 |
| ArcFace [0.2, 0.0] | 0.00 | -57.70 | -140.07 | -4.72 | 0.00 | -7.75 | 75.31 | 92.72 | 0.00 | -10.44 | 64.32 | 86.37 |
| ArcFace [0.2, 0.2] | 0.00 | -4.56 | 27.91 | 70.28 | 0.00 | -3.55 | 72.94 | 57.76 | 0.00 | -2.86 | 43.79 | 52.27 |
| ArcFace [0.4, 0.2] | 0.00 | -6.31 | 66.22 | 13.25 | 0.00 | -3.49 | 76.69 | 52.72 | 0.00 | -3.90 | 56.85 | 91.29 |
| ArcFace [0.4, 0.4] | 0.00 | -0.78 | 89.23 | -8.47 | 0.00 | -0.97 | 89.53 | -108.01 | 0.00 | -1.57 | 78.34 | 20.47 |
| Dlib [0.0, 0.0] | 0.00 | 18.22 | 1.56 | 65.46 | 0.00 | -57.38 | -18.55 | -3.43 | 0.00 | 88.05 | -245.95 | 17.61 |
| Dlib [0.2, 0.0] | 0.00 | -58.50 | -95.79 | 85.62 | 0.00 | -7.66 | 84.90 | 89.53 | 0.00 | -10.44 | 71.30 | 90.42 |
| Dlib [0.2, 0.2] | 0.00 | -4.43 | 44.49 | 93.54 | 0.00 | -3.53 | 78.78 | 74.80 | 0.00 | -2.80 | 57.89 | 40.54 |
| Dlib [0.4, 0.2] | 0.00 | -6.15 | 75.03 | 65.59 | 0.00 | -3.39 | 83.58 | 81.78 | 0.00 | -3.82 | 66.59 | 80.25 |
| Dlib [0.4, 0.4] | 0.00 | -0.71 | 99.41 | -24.07 | 0.00 | -0.92 | 89.94 | -8.80 | 0.00 | -1.54 | 89.22 | 81.15 |
| SFace [0.0, 0.0] | 0.00 | 18.22 | -10.87 | 68.87 | 0.00 | -57.38 | 1.61 | 4.85 | 0.00 | 88.05 | -232.48 | -16.46 |
| SFace [0.2, 0.0] | 0.00 | -57.31 | -104.91 | 73.84 | 0.00 | -7.79 | 86.78 | 64.75 | 0.00 | -10.44 | 70.66 | 47.12 |
| SFace [0.2, 0.2] | 0.00 | -4.45 | 39.93 | 30.68 | 0.00 | -3.67 | 89.76 | 53.18 | 0.00 | -2.92 | 69.34 | 7.82 |
| SFace [0.4, 0.2] | 0.00 | -6.24 | 71.34 | 4.66 | 0.00 | -3.47 | 95.32 | -9.55 | 0.00 | -3.94 | 73.06 | -21.85 |
| SFace [0.4, 0.4] | 0.00 | -0.78 | 95.67 | 59.82 | 0.00 | -0.98 | 72.82 | -124.64 | 0.00 | -1.60 | 74.44 | -10.49 |

### D.4 EXPERIMENTS ON COMPAS WITH THREE-CLASS SENSITIVE ATTRIBUTES

We experiment with three categories of sensitive attributes: black, white, and others, and show the result in Table 11. Note EOp is not defined in the case with more than two categories. Table 11 shows our proposed framework with global estimates is consistently and significantly better than the baselines, which is also consistent with the results from Table 1.

Table 11: Normalized estimation error on COMPAS. Each row is a different target model $f$.

| COMPAS | DP Normalized Error (%) ↓ | | | | EOd Normalized Error (%) ↓ | | | |
|---|---|---|---|---|---|---|---|---|
| True disparity: ~ 0.2 | Base | Soft | Global | Local | Base | Soft | Global | Local |
| tree | 24.87 | 64.91 | **13.84** | 25.16 | 30.15 | 67.30 | **13.11** | 27.84 |
| forest | 23.94 | 63.61 | **10.00** | 26.64 | 29.19 | 65.73 | **11.61** | 33.85 |
| boosting | 24.87 | 64.91 | **13.84** | 25.44 | 30.15 | 67.30 | **15.74** | 33.20 |
| SVM | 40.37 | 66.97 | **25.96** | 34.73 | 49.57 | 70.58 | **29.33** | 42.91 |
| logit | 16.71 | 64.18 | **7.39** | 22.17 | 17.23 | 66.47 | **7.02** | 25.38 |
| nn | 18.60 | 62.92 | **5.38** | 16.58 | 22.91 | 65.51 | **5.90** | 22.63 |
| compas_score | 29.00 | 63.33 | **10.02** | 31.32 | 33.43 | 65.82 | **12.15** | 36.03 |

### D.5 PRELIMINARY RESULTS ON DISPARITY MITIGATION WITH OUR CALIBRATION FRAMEWORK

We apply our calibration framework to mitigate disparity during training. Specifically, the local method is applied on the CelebA dataset. The preprocess of the dataset and generation of noisy sensitive attributes are the same as the experiments in Table 2. The backbone network is ViT-B_8 (Dosovitskiy et al., 2020). The aim is to improve the classification accuracy while ensuring DP, where $\widehat{\Delta}(\widetilde{D}, f) = 0$ is the constraint during training. Specifically, the optimization problem is

$$\min_{f} \quad \sum_{n=1}^{N} \ell(f(x_n), y_n)$$
$$s.t. \quad \widehat{\Delta}(\widetilde{D}, f) = 0,$$

where $\ell$ is the cross-entropy loss. Recall $\widehat{\Delta}(\widetilde{D}, f)$ is obtained from our Algorithm 1 (Line 8), and $\widetilde{D} := \{(x_n, y_n, \tilde{a}_n) | n \in [N]\}$. Noting the constraint is not differentiable since it depends on the sample counts, i.e.,

$$\widetilde{H}[\tilde{a}, k] = \mathbb{P}(f(X) = k | \widetilde{A} = \tilde{a}) \approx \frac{1}{N} \sum_{n=1}^{N} \mathbb{1}(f(x_n = k | \tilde{a}_n = \tilde{a})).$$

To make it differentiable, we use a relaxed measure (Madras et al., 2018; Wang et al., 2022) as follows:

$$\widetilde{H}[\tilde{a}, k] = \mathbb{P}(f(X) = k | \widetilde{A} = \tilde{a}) \approx \frac{1}{N_{\tilde{a}}} \sum_{n=1, \tilde{a}_n=\tilde{a}}^{N} \boldsymbol{f}_{x_n}[k],$$

where $\boldsymbol{f}_{x_n}[k]$ is the model's prediction probability on class $k$, and $N_{\tilde{a}}$ is the number of samples that have noisy attribute $\tilde{a}$. The standard method of multipliers is employed to train with constraints (Boyd et al., 2011). We train the model for 20 epochs with a stepsize of 256. Table 12 shows the accuracy and DP disparity on the test data averaged with results from the last 5 epochs of training. From the table, we conclude that, with any selected pre-trained model, the mitigation based on our calibration results significantly outperforms the direct mitigation with noisy attributes in terms of both accuracy improvement and disparity mitigation.

Table 12: Disparity mitigation with our calibration framework. Results are averaged with results from the last 5 epochs. DP is the considered fairness metric. Base: Direct mitigation with noisy sensitive attributes. Facenet, Facenet 512, *etc.*: Pre-trained models to generate feature representations used to simulate the other two auxiliary models.

| CelebA | $\Delta^{\text{DP}}(D^{\text{text}}, f) \downarrow$ | Accuracy $\uparrow$ |
|--------|------|------|
| Base | 0.0578 | 0.8422 |
| Facenet | 0.0453 | 0.8466 |
| Facenet512 | 0.0273 | 0.8557 |
| OpenFace | 0.0153 | 0.8600 |
| ArcFace | 0.0435 | 0.8491 |
| Dlib | 0.0265 | 0.8522 |
| SFace | 0.0315 | 0.8568 |

