# OpenReview forum: "Evaluating Fairness Without Sensitive Attributes: A Framework Using Only Auxiliary Models"
_ICLR.cc/2023/Conference — Submitted to ICLR 2023_

### Official Review · Reviewer_pHkG · 2022-10-23

**Confidence:** 2
**Clarity, Quality, Novelty And Reproducibility:** Some typos, e.g., M & K, makes it dif…
**Correctness:** 4
**Technical Novelty And Significance:** 3
**Empirical Novelty And Significance:** 3
**Recommendation:** 6

**Strength And Weaknesses:**

Strength:

+ It proves the proposed method has less error than the direct estimate model under a threshold.
+ The proposed method is a plug-in method, it can be used as long as the auxiliary model’s input features have overlap with the target features.

Weakness:

- This paper removes some assumptions but the common assumption on T is still very strong.
- The proposed research is evaluated on experiments with binary classification and binary sensitive attributes.
- This work estimates the unfairness metrics in the missing sensitive attribute setting. As a framework, it would be more convincing if the mitigation solution is also provided.


**Summary Of The Paper:**

This proposed a framework to evaluate fairness when ground-truth sensitive attributes are not accessible. It calibrates the noisy fairness metrics using auxiliary models only and it drops many assumptions. The auxiliary model is flexible, the auxiliary model can be used if the input features have some overlaps with targets.

**Summary Of The Review:**

This work proposes a new method with fewer assumptions to estimate fairness metrics in the setting of missing sensitive attributes using auxiliary models. The theoretical analytics and the experimental results show the effectiveness. The proposed method is a plug-in method which means it is flexible.

It would be more complete and convincing if the estimated metrics are used for bias mitigation.

---

> ### Author Response · Authors · 2022-11-11
> **Response to Reviewer pHkG**
>
>
> **C1. The common assumption on T is still very strong**
>
> If we understand it correctly, it seems the reviewer thinks we make certain assumptions on $\bf T$. If this is the case, we would like to clarify that we do drop the common assumption on $\bf T$ in our method, which is one of our contribution. To avoid confusion, we revised the paragraph above Section 3 to highlight this point: "Many prior works (Awasthi et al., 2021; Prost et al., 2021; Fogliato et al., 2020) assume $\widetilde A$ and $f(X)$ are conditionally independent on $A$. We drop this assumption in our theoretical framework." Please feel free to let us know if we misunderstand it or which part of the paper makes the reviewer believe we have strong assumption on $\bf T$.
>
>
> **C2. The proposed research is evaluated on experiments with binary classification and binary sensitive attributes.**
>
> Thanks for the comment. We have added an experiment (Table 11, Appendix D.4, Page 29) when considering three sensitive attributes (black, white, others) in COMPAS. The other settings are the same as Table 1. Table 11 shows our method with global estimates is consistently and significantly better than baselines.
>
> **C3. Mitigation**
>
> We thank the reviewer for the suggestion. First, we would like to emphasize that we believe we have made it perfectly clear that our scope is limited to measuring fairness rather than mitigating bias, as we repeatedly mention it in introduction, related work, and most obviously, the title (``Evaluating Fairness XXX''). We believe we do not overclaim anything in the paper, and measuring fairness without mitigation itself is a great challenge. And in many real-world scenarios, estimating fairness is a valuable task *per se*.
>
>
> However, we understand that without mitigation it *might be* incomplete. To show our method can be applied to mitigation, we run an in-processing mitigation algorithm based on our framework. Specifically, we use the local method in CelebA dataset. We want to improve the classification accuracy while ensuring DP, *i.e.* $\widehat \Delta_{\textsf{DP}}(\widetilde D,f) = 0$ is the constraint during training. Recall $\widehat \Delta_{\textsf{DP}}(\widetilde D,f)$ is obtained from our Algorithm 1 (Line 8).
> We use the standard method of multipliers to optimize the training with constraints [1]. See more details about the design of the mitigation algorithm in Appendix D.5 (Page 29).
> Table 12 (Page 30) shows the mitigation based on our calibration results significantly outperform the direct mitigation with noisy attributes in terms of both accuracy improvements and disparity mitigation.
>
> [1] Boyd, S., Parikh, N., Chu, E., Peleato, B. and Eckstein, J. Distributed optimiza-
> tion and statistical learning via the alternating direction method of multipliers. Foundations and
> Trends in Machine learning, 3(1):1–122, 2011.
>
>
> **C4. Typos**  $M$ & $K$
>
> We consider a problem with $M$-class sensitive attributes and $K$-class labels as defined in the first paragraph of Section 2 (Page 2). We appreciate it if the reviewers could be more specific about the detailed place of typos.

---

> > ### Comment · Reviewer_pHkG · 2022-12-03
> > **reply to the authors' response**
> >
> > I appreciate the authors' responses. It resolved my concerns about the T assumption and the experiment.

---

### Official Review · Reviewer_Za6Z · 2022-10-25

**Confidence:** 4
**Correctness:** 4
**Technical Novelty And Significance:** 3
**Empirical Novelty And Significance:** 2
**Recommendation:** 6

**Clarity, Quality, Novelty And Reproducibility:**

The paper is clearly written, and the theoretical results are interesting. However, as explained above, the proposed framework and experimental results are relatively less novel.

**Strength And Weaknesses:**

[Strength]

S1: The paper tries to evaluate the fairness of the models without accessing sensitive attributes, which is an important and practical issue.

S2: The theoretical analyses in the paper give several nice insights. For example, I enjoy reading Section 3, which well analyzes the factors that affect the estimation error in model fairness.

S3: The proposed algorithm is effective in specific empirical scenarios, as shown in their theoretical results.

[Weakness]

Overall, although the paper has various interesting aspects in its theoretical discussion, the proposed framework and experimental results are relatively weak.

W1: The paper uses HOC as their estimator without comparing other methods. Although the paper explains some reasons for choosing HOC (e.g., free of training), it is not clear that the proposed framework can show similar results when using other estimators. The experimental results seem to be somewhat determined by the performances of HOC, so it may be questionable whether the current experimental results are dependent on HOC or not. It would be great if the paper can clarify the robustness of the proposed framework w.r.t. the estimators.

W2: As shown in Tables 1–3, the performances when using global T and local T_k are very different, but there is no detailed discussion on how to choose either global or local. This result makes the framework less attractive, as the estimation errors are largely dependent on a specific hyperparameter (i.e., choosing global or local).

**Summary Of The Paper:**

The paper aims to evaluate the fairness of the model without accessing sensitive group attributes. The main idea is to utilize auxiliary models of estimating group attributes by calibrating their outputs. The paper first theoretically explains why we need calibration on the auxiliary models. Then, the paper suggests a calibration-based training framework that uses the existing distribution estimation techniques. The proposed algorithm is evaluated in two datasets, COMPAS and CelebA, and achieves a more precise evaluation when the auxiliary model is accurate and the target model is highly biased.

**Summary Of The Review:**

The paper tries to solve the important issue, which evaluates model fairness in a real-world scenario where sensitive attributes are not available. As the paper gives several insights in their theoretical discussion, I vote for marginally accepting this paper if it can clarify the contribution of the proposed framework and enhance the experimental results (or discussion).

---

> ### Author Response · Authors · 2022-11-11
> **Response to Reviewer Za6Z**
>
> **C1. Performance with other estimators**
>
> We thank the reviewer for the comment. We understand it is a reasonable concern since we claim our method is a general framework that can plug in any existing estimator while we only show the experimental results when using HOC. And therefore it is unclear how strongly our framework is coupled with HOC.
>
> *First*, we admit the experimental results we have shown are dependent on HOC. Although our framework is compatible with any estimator, we do not claim that the performance of the framework would be independent of the estimator. If one uses a bad estimator, the results from our framework will be bad as well (simply garbage in garbage out). We hope this is clear.
>
> *Second*, the reason why we choose HOC is because it can estimate transition matrices more accurately than other methods in noisy label literature. As we prove in Theorem 3, if all transition matrices are perfectly estimated, then our framework would output the perfectly accurate fairness measures. In practice, performance of estimator could be the bottleneck of the framework's accuracy. HOC is the most accurate estimator that we are aware of. We also tried other estimators, *e.g.,* [1--2], but they highly depend on training separate models, which are likely to overfit the training data and converge to the auxiliary model. As a result, it will return a trivial solution where $\bf T$ is almost an identity matrix. Using an identity matrix in our framework is equivalent to no calibration (the Base method). For example, adapting [1] returns the following normalized error ($\times 100$):
> | Method      | DP Global   | DP Local | EOd Global | EOd Local | EOp Global | EOp Local |
> | ----------- | ----------- | ----------- | ----------- | ----------- | ----------- | ----------- |
> | Base        | 15.33       |  /       | 4.11       |     /     | 2.82       | /         |
> | [1]         | 15.37       | 15.49   | 4.07      | 4.02     | 2.86       | 2.95       |
>
> which is very close to Base. We have included related discussions in Appendix C.2.
>
> If the reviewer has any suggestion on estimator, please let us know. In reality, using a better estimator can improve the performance of our framework. Since our framework's design is perfectly compatible with all estimators, as the noisy label community is designing better estimators, our framework can also benefit.
> We will clarify it in the paper.
>
>
> **C2. Choice between global T and local T**
>
>
> We thank the reviewer for the question. We have discussed how to choose global or local in the **Takeaways** paragraph at the bottom of Page 8 in our original submission. In short, we can choose local when the auxiliary model is accurate
> and global otherwise. We also have more detailed discussions on why we can make this choice in the paragraph **Local vs. Global** on Page 26 in Appendix D.3.
>
>
>
> [1] Northcutt, C., Jiang, L. and Chuang, I. Confident learning: Estimating uncertainty in dataset labels. JAIR 2021.
>
> [2] Xia, X., Liu, T., Wang, N., Han, B., Gong, C., Niu, G. and Sugiyama, M. Are anchor points really indispensable in label-noise learning?. NeurIPS 2019.

---

> > ### Comment · Reviewer_Za6Z · 2022-11-18
> > **Response to Authors**
> >
> > I appreciate the authors' responses that contain honest explanations.
> >
> > After reading the responses, I still think the weakest part of this work might be the dependency on the estimator and hyperparameters. The reason that I keep concerned about this aspect is that this work aims to solve a practical research topic (estimating fairness without group attributes). However, such dependent characteristics may make the proposed solution less practical.
> >
> > Nevertheless, since I believe the theoretical parts give some good insights, I will vote for this paper to be marginally above the acceptance threshold. I hope to hear from other reviewers during the discussion period.

---

> > > ### Author Response · Authors · 2022-11-22
> > > **Thanks for supporting our paper & more discussions on HOC dependence**
> > >
> > > We thank the reviewer for supporting our paper. The dependence on the HOC estimator is indeed a valid concern in practice. We would like to discuss it more from the following four aspects.
> > >
> > >
> > > - **Our work is a first step of solving a technically challenging problem under a fresh view rather than a final step of proposing a perfect solution.**
> > > As can be sensed intuitively, evaluating and mitigating unfairness without ground-truth sensitive attributes is a technically challenging problem because the sensitive attribute is such vital information in studying fairness. However, this is a practical problem that we need to face often in practice (a problem that the industry, like Google and Meta, is frequently facing), as we already motivated in the Introduction. We do not claim we fully solve such a hard problem; instead, we provide a new view to the fairness community. Specifically speaking, our work aims to bring more attention to solving the problem by estimating the transition matrix, a concept in noisy label literature that the fairness community has overlooked. Using our framework, we can turn the problem into a transition matrix estimation problem, which is well-studied in the noisy label literature. We believe this is an important step that opens more possibilities for future work along this line and contributes to the development of both fairness and noisy label communities.
> > >
> > >
> > > - **Using HOC gives good performance and does not have major disadvantages in practice.**
> > > We admit current solutions are not perfect and their effectiveness may largely depend on HOC. But experimental results show using HOC together with our framework can beat the baselines. In practice, even if practitioners do not choose to flexibly plug in other estimators, and rather just stick to using HOC, they can get decent results. Using HOC is not something that has major limitations in practice. In fact, the assumptions in HOC are mild and the algorithm is easy to implement and run.
> > >
> > >
> > > - **The estimator bottleneck comes from the noisy learning literature rather than our technical contribution.**
> > > In our framework, if the estimator is perfect (*i.e.* ${\bf T}_k = \widehat {\bf T}_k$ and ${\bf p} = \hat{\bf p}$), then it is guaranteed that our fairness estimation is *100\% accurate*. However, needless to say, no estimator is perfect. Any estimation algorithm has errors, certainly in our case, trying to estimate sensitive attributes without their ground-truth information is a challenging task and the estimators cannot be perfect. To the best of our knowledge, HOC is the existing estimator that fits our framework best. This is the limitation in the noisy label literature that is independent of our technical contribution.
> > >
> > >
> > > - **The estimator could be improved in the future.**
> > > We can improve estimator by considering more information, *e.g.* internal relationship between ${\bf T}_k = \widehat {\bf T}_k, \forall k$ and ${\bf p} = \hat{\bf p}$. Besides, it is also possible to get rid of estimating the above statistics by borrowing other ideas from the noisy label literature, such as peer loss [1]. Those are examples of the potential future works that our work can inspire. We provide a framework that connects noisy labels and fairness, and many future works can be devoted to improving technical aspects. As the noisy label community improves its techniques, our framework can benefit.
> > >
> > >
> > > [1] Liu, Yang, and Hongyi Guo. "Peer loss functions: Learning from noisy labels without knowing noise rates." International conference on machine learning. PMLR, 2020.

---

### Official Review · Reviewer_vWhJ · 2022-10-26

**Confidence:** 2
**Correctness:** 2
**Technical Novelty And Significance:** 2
**Empirical Novelty And Significance:** 2
**Recommendation:** 5

**Clarity, Quality, Novelty And Reproducibility:**

Suffers from clarity.

Some novelty in reducing the dependence on assumptions for the auxiliary models.

**Strength And Weaknesses:**

An important problem is studied, motivation from real-world settings is lacking.
Clarity in writing can be improved.

**Summary Of The Paper:**

The paper focuses on learning the sensitive attributes using auxiliary models for estimating fairness metrics. Theoretical analysis is done to understand the relationship between the directly measured fairness metrics and their corresponding ground-truth metrics.

**Summary Of The Review:**

The paper suffers from clarity, and the writing could be improved for better understanding.
The problem studied is of importance but would be helpful to add motivation with some problems from real-world datasets or settings.

---

> ### Author Response · Authors · 2022-11-11
> **Response to Reviewer vWhJ**
>
> **Motivation with some problems from real-world datasets**
>
> The problem motivation comes from the practical situation that the sensitive attribute information is often unavailable in a machine learning system (because it contains sensitive demographic information) due to privacy regulation, most notably GDPR. This is a practical concern that industry is facing, *e.g.* Google and Meta. Given those facts, the problem is well-motivated, as we explain in details in the first paragraphs of Introduction. In addition, our theoretical results (Theorem 1 and its analyses) show that directly using classifiers to label sensitive attributes would lead to a large error and therefore we propose the calibration algorithm. Our numerical results on real-world datasets (COMPAS, CelebA) also show the fairness metric will be wrongly measured without proper calibration. Besides, in the revised version, we add Table 12 (Page 30) to numerically show the harm of directly mitigating the disparity using noisy attributes and the necessity of mitigation relying on our calibration framework on real-world datasets.

---

### Official Review · Reviewer_oWCf · 2022-10-27

**Confidence:** 3
**Correctness:** 4
**Technical Novelty And Significance:** 2
**Empirical Novelty And Significance:** 3
**Recommendation:** 3

**Clarity, Quality, Novelty And Reproducibility:**

Clarity: The clarity could be significantly improved. I'd suggest making the paper less dense by simplifying more often to the binary case, spend more time explaining the core intuition, clarify the insight beyond prior work, and include a description of critical dependencies like the HOC algorithm.

Quality: Overall the work seems thorough and high quality.

Novetly: Because one of the two core methods (Global) seems equivalent to past work (relying on the conditional indepence assumption) and the other is not well described (HOC) the novelty relative to past work seems likely there but not well articulated.

Reproducibility: I do not believe the code is open sourced but the work seems generally reproducible.


**Details Of Ethics Concerns:**

See above on including a discussion of the risks of predicting group information.


**Strength And Weaknesses:**

S1. Technically challenging and important problem.

S2. The formulation provided is general and clarifies what information is needed for accurate estimation.

S3. The experiments provide a useful breakdown of when different methods excel.

W1. Clarity: While the formulation is general, it is hard to parse in many cases the intuition behind many terms.  Further, a core contribution of the paper seems to be about leveraging noisy label approaches most significantly the HOC algorithm, but the algorithm is not at all described in the paper.

W2. One core claim of the paper is that they don't need the conditional independence assumption of past literature but the "Global" method (which often works best at least for COMPAS) seems to directly be enforcing the conditional independence assumption of past work.  As such, it is not clear what is the core contribution of the paper.

W3. Strongly strongly suggest a discussion of the limitation and ethical fraught-ness of using classifiers for sensitive attributes (eg predicting race based on name or gender based on an image).  This research is still important but there are many caveats with predicting these attributes in practice.


Details while reading:

Not obvious how Thm 1 would be extended to equality of odds/opportunity.

For clarity purposes overall I think the paper would be easier to read assuming binary labels and groups, with generalizations to many groups being included later.

Sec 4.1 - Wouldn't use T to estimate all $T_k$ be exactly the conditional independence assumption from past work as described in Sec 2 (Transition Matrix)?

Also how would $T_k$ be estimated based on $\tilde{D}$? I thought $T_k$ needs both $A$ and $\tilde{A}$ as it is the transition probabilities between the two.

HOC seems critical to the method in this paper but is not described.  For completeness please include



**Summary Of The Paper:**

In this paper the authors study how to compute different group fairness measures when relying on pre-trained models to predict group attributes.  The paper provides a general mathematical formulation that focuses on estimating the transition probabilities (essentially the normalized confusion matrix) between ground truth and predicted group information conditioned on main task class.  The paper then suggests multiple approaches for this estimation, including one from the noisy label literature, and empirically tests which methods work best.  They find that in some cases (some datasets and fairness goals) their method outperforms baselines.


**Summary Of The Review:**

Overall the paper is a thorough study of an important problem, but the key contributions that move the field forward are hard to extract.

---

> ### Author Response · Authors · 2022-11-11
> **Response to Reviewer oWCf (Part 1)**
>
> **C1. Clarity: hard to parse in many cases the intuition behind many terms.**
>
> We thank the reviewer for the comment. In our writing, we try our best to explain the meaning behind the mathematical terms. For example, we have listed term-by-term the intuition in Theorem 1 (Page 4). It would be more helpful if the reviewer can point out specifically where we are not clear enough so that we can clarify them for the readers.
>
> **C2. HOC algorithm.**
>
> We thank the reviewer for raising the concern. However, we would like to re-emphasize (as we repeatedly mentioned in the paper) that HOC is **not** our key contribution. What we propose is a general framework that can be used by plugging in any transition matrix estimator proposed in the noisy label area. And HOC is just one such estimator that we choose to demonstrate the effectiveness of our algorithm because it has been shown accurate in estimating transition matrix by the noisy label literature. Technically speaking, any estimator can be used. However we understand the possible confusion without knowing HOC in detail. To this end, we add a section in Appendix C.1 on Page 24 of the revised version to explain how HOC works. Hope this will resolve the reviewer's concern.
>
> **C3. Global method and the conditional independence assumption vs past work**
>
> We thank the reviewer for pointing it out. We understand why the reviewer has this question. This is a technical confusion that requires in-depth explanations. In short, using global estimates $\widehat {\bf{T}}$ is only a heuristic in experiments to estimate $\bf T_k$. Our method does not rely on the conditional independence assumption.
>
> Please note the following  notation differences: probabilities $\bf T_k$, $\bf T$ and probability estimates $\widehat{\bf T}_k$, $\widehat{\bf T}$. In practice, due to **the existence of estimation errors** when estimating transition matrix from data that is independent of the conditional independence assumption, the local estimates $\bf T_k \approx \widehat {\bf T}_k$ may not always be the best choice. Using global $\widehat{\bf T}$ to approximate local $\bf T_k$, i.e., $\bf T_k \approx \widehat{\bf T}$, is just a simple heuristic that we use to stabilize numerical computation of running estimator in practice. We only *hope* the global estimate $\widehat {\bf T}$ is closer to $\bf T_k$ than the local estimate $\widehat {\bf T}_k$. We agree that the usage of global estimates can be supported if the conditional independence holds. However, its benefit also depends on how inaccurate $\widehat{\bf T}_k$s are. Therefore, using global estimates is not equivalent to assuming the condition independence and we always want to get $\bf T_k$ in our method.
>
> We understand this is a great point to clarify. To avoid confusion, we have revised the paper to focus on $\bf T_k$ in our theorems and algorithms (before Section 5) and talk about the global estimates only as a heuristic in practical implementation of $\bf T_k$ (Section 5.1 Experiment Setup. Practical Estimates of $\bf T_k$: Local vs. Global).
>
> **C4. Core contribution of the paper**
>
> It seems the reviewer questions our contribution because we *seem* to assume conditional independence. If that is the case, we respectfully disagree for two main reasons.
>
> *First*, we hope it is now clear from the response to C3 that we **do not** assume conditional independence.
>
> *Second*, even if we do, it does not nullify our methodological contribution. Specifically, in addition to dropping the conditional independence assumption, we also drop other commonly made assumptions:  1) access to labeling resource, 2) access to auxiliary model’s training data, 3) data i.i.d.
>
> *Finally*, if the reviewer is still unclear about our contribution, please see summary in the bulleted items on Page 2, including both theoretical contributions (Theorems 1--3, Corollaries 1--2) and algorithmic contributions (Algorithms 1--2).
>
> **C5. Global method works well on COMPAS**
>
> If we understand it correctly, the reviewer asks why the global method works well on COMPAS. We find the global method works well if $\bf T$ cannot be stably estimated (as shown in the COMPAS experiments), and the local method works well if $\bf T_k$s are estimated accurately (CelebA experiments). This is reasonable as we have proved that the performance of our method depends on the transition matrices (Theorem 3, the $\varepsilon$ term).

---

> > ### Author Response · Authors · 2022-12-03
> > **gentle follow up**
> >
> > Dear Reviewer oWCF,
> >
> > Thank you again for your review comments. We have provided detailed responses. We'd like to follow up to see if there is anything else that we could further clarify. Thank you.
> >
> > Best,
> >
> > Authors

---

> ### Author Response · Authors · 2022-11-11
> **Response to Reviewer oWCf (Part 2)**
>
>
> **C6. Limitation and ethical fraught-ness of using classifiers for sensitive attributes**
>
> We thank the reviewer for the comment. We do understand this is a sensitive topic. However we want to point out the fact that we are not the first to use auxiliary models to estimate fairness. In fact, this is a standard practice in industry used by Google (Awasthi et al., 2021), Meta (Alao et al., 2021) as well as numerous works (Elliott et al., 2009; Diana et al., 2022) (we highlighted them at the bottom of Page 1). We have also included an ethical statement on Page 10 to discuss the limitations.
>
> We could potentially add two more points in the ethical statement. *First*, from a privacy aspect, using an imperfect classifier actually protects the user privacy. It has been proved in [1] that using noisy attributes (labels in [1]) has differential privacy guarantee. *Second*, from a fairness aspect, many of our key methods and contributions are designed to reduce the potential risk of using auxiliary models to estimate fairness directly, as it might lead to a radically wrong estimate. For example, our Theorem 1 shows that directly using auxiliary models without calibration would lead to a large estimate error, and our method is proposed to address it (**zero** error with true $\bf T_k$ as proved in Theorem 3). In addition, we do not make conditional independence assumptions in the existing literature since we cannot control it in practice, which might lead to a unreliable estimate.
>
>
> **C7. Not obvious how Thm 1 would be extended to equality of odds/opportunity.**
>
> We thank the reviewer for the question. However we do include both theoretical and experimental results on Equal Opportunity and Equal Odds in the paper. As explained at the bottom of Page 2, ``To save space, all our discussions in the main paper are specific to DP. We include the complete derivations for EOd and EOp in the Appendix.''
> Please check the full version of Theorem 1 for results wrt DP, EOd, and EOp at the bottom of Page 16.
>
> **C8. Assuming binary labels and groups at first**
>
>
> We thank the reviewer for the suggestion. However the reason why we choose multi-class labels/groups and use a matrix form is because it is *more* concise, not less. For example, if we represent Theorem 2 using a binary example, then it would be as cumbersome as the following:
> $$
> H[1,k] = \frac{\left(1-e\_2^k \right)   \tilde p\_1 \cdot  \widetilde H[1,k] - e\_2^k  \tilde p\_2 \cdot \widetilde H[2,k]}{\left(1-e\_1^k-e\_2^k\right)p_1},
> $$
> and
> $$
> H[2,k] = \frac{ \left(1-e_1^k\right)\tilde p_2\cdot  \widetilde H[2,k] - e_1^k \tilde p_1 \cdot  \widetilde H[1,k] }{\left(1-e_1^k-e_2^k\right)p_2},
> $$
> where $e_1^k := \mathbb(\widetilde A = 2 | A=1, f(X)=k)$, $e_2^k := \mathbb P(\widetilde A = 1 | A=2, f(X)=k)$, ${\bf p} = [p_1, p_2]^\top$, and $\tilde{\bf p}=[\tilde p_1, \tilde p_2]$. In a matrix form, we can simply let $K=2$ to reproduce the above result.
>
>
> We hope the above example is clear enough to show that it would not be neater had we used a binary form. Besides, we also discussed some special binary case as in Corollary 1 and Corollary 2.
>
>
> **C9. Estimate T without A**
>
> If we understand it correctly, the reviewer asks how we estimate $T$ without $A$. In fact, this is exactly the intuition behind HOC and how we adapt HOC. We have explained it in Section 4.3 (1st paragraph). In addition, we include more details about HOC in Appendix C.1 (Page 24). In short, we have multiple auxiliary models and we analyze the agreements and disagreements among them to estimate $\bf T$. These high-order information brings us additional equations to facilitate the estimates of these hidden parameters (transition models). This line of approaches are theoretically guaranteed to uniquely recover the true $\bf T$ when Assumptions 1 and 2 hold.
>
> [1] Ghazi, B., Golowich, N., Kumar, R., Manurangsi, P. and Zhang, C. Deep learning with label differential privacy. NeurIPS 2021.
>
> [2] Zhu, Z., Song, Y., Liu, Y. Clusterability as an alternative to anchor points when
> learning with noisy labels. ICML 2021.

---

### Author Response · Authors · 2022-11-11
**General response to all reviewers and ACs**

We thank all the reviewers and ACs for their detailed and helpful comments. We have carefully addressed all the comments and revised our paper and highlighted major revisions in blue. Particularly, we have highlighted that our proposed theoretical framework does not rely on assuming conditional independence of the noise transition matrix, and added more experiments to show our method works well in the case with three-class sensitive attributes (Table 11) and disparity mitigation (Table 12).

We have responded in detail to each reviewer individually. The revised paper has been uploaded. Please feel free to let us know if there is still any confusion.

---

### Decision · Program_Chairs · 2023-01-20

**Decision:**

Reject

**Justification For Why Not Higher Score:**

More experimentation with other estimators is recommended.

**Justification For Why Not Lower Score:**

N/A

**Metareview: Summary, Strengths And Weaknesses:**

The paucity of reliable data and privacy issues may render some attributes of nodes in graph data inaccessible, which is a challenge for fair graph learning. Assumptions of (conditional) independence are avoided, and a framework that uses only off-the-shelf auxiliary models is developed. One of the key issues addressed is how to reduce the negative impact of imperfectly-predicted (sensitive) attributes on the fairness metric, without knowing the ground-truth about the sensitive attribute values. The “noisy label learning” literature is leveraged to derive bounds relating the measured fairness metrics and the corresponding ground-truth metrics. Some key statistics are estimated as a result, and along with the derived bounds, are used to calibrate the fairness metrics.

The work seems thorough. Among other concerns, it would be good to experiment with many estimators in addition to HOC in order to clarify how well the method does in practice. The authors are also encouraged to decrease the clutter and focus on the binary case.